# Accuracy of Structure-from-Motion/Multiview Stereo Terrain Models: A Practical Assessment for Applications in Field Geology

**Terry L. Pavlis * and Laura F. Serpa**

Department of Earth, Environmental and Resource Science, The University of Texas at El Paso, El Paso, TX 79968, USA; lfserpa@utep.edu
* Correspondence: tlpavlis@utep.edu

**Abstract:** We assess the accuracy of Structure-from-Motion/Multiview stereo (SM) terrain models acquired ad hoc or without high-resolution ground control to analyze their usage as a base for inexpensive 3D bedrock geologic mapping. Our focus is on techniques that can be utilized in field projects without the use of heavy and/or expensive equipment or the placement of ground control in logistically challenging sites (e.g., steep cliff faces or remote settings). We use a Terrestrial Light Detection and Ranging (LiDAR) survey as a basis for the comparison of two types of SM models: (1) models developed from images acquired in a chartered airplane flight with ground control referenced by natural objects located on Google Earth scenes; and (2) drone flights with a georeference established solely from camera positions located by conventional, differentially corrected Global Navigation Satellite systems (GNSS). We find that all our SM models are indistinguishable in scale from the LiDAR reference model. The SM models do, however, show rigid body translations and rotations, with translations generally within the 1–5 m size of the natural objects used for ground control, the resolution of the GNSS receivers, or both. The rigid body rotations can be attributed to a poor imaging plan, which can be avoided with survey planning. Analyses of point densities in various models show a limitation of Terrestrial LiDAR point clouds as a mapping base due to the rapid falloff of resolution with distance. In contrast, SM models are characterized by relatively uniform point densities controlled by camera optics, the numbers of images, and the distance from the target. This uniform density is the product of the Multiview stereo step in SM processing that fills areas between key points and is important for bedrock geologic mapping because it affords direct interpretation on a point cloud at a relatively uniform scale throughout a model. Our results indicate that these simple methods allow SM model construction to be accurate to the range of conventional GNSS with resolutions to the submeter, even cm, scale depending on data acquisition parameters. Thus, SM models can, and should, serve as a base for high-resolution geologic mapping, particularly in a steep terrain where conventional techniques fail. Our SM models appear to provide accurate visualizations of geologic features over km scales that allow detailed geologic mapping in 3D with a relative accuracy to the decimeter or centimeter level and absolute positioning in the 2–5 m precision of GNSS; a geometric precision that will allow unprecedented new studies of any geologic system where geometry is the fundamental data.

**Keywords:** Structure-from-Motion; 3D mapping; accuracy assessment

## 1. Introduction

Geologic field observations are a basic tool in the geosciences, but common methods for making these observations have changed very little since the early 20th century when aviation led to the first aerial photography and travel on horseback was replaced by automobiles and aviation e.g., [1]. A typical geology field study requires a trained observer to physically examine geologic features at a wide range of scales (cm to km, typically),

make measurements in the field, collect samples for additional laboratory examination, identify the locations of all observations, and compile these data in some way (map, interpretative figures, etc.) that can be shared with the broad community of geoscience researchers and the general public. In the process, a field geologist may hike across difficult terrain with logistical obstacles ranging from weather to fending off an occasional predator while making their detailed observations. Typically, aviation has been limited to obtaining remotely sensed data over large areas (e.g., aerial photography or satellite imagery), but in remote areas, it might also be used for access and aerial reconnaissance. When these data are released as a publication, they are typically shown as a few 2D representations (i.e., maps and cross-sections) that are assumed to provide sufficient information for most current applications. This assumption is rarely true. In particular, most field data are inherently three-dimensional (3D) and 2D representations of these data can be misleading.

Fortunately, new tools are available that begin to solve this 3D visualization problem. Specifically, 3D terrain models are revolutionizing field geology by allowing visualization of complex and/or remote geologic features where conventional 2D maps and vertical aerial photography distort or obscure geologic observations. Three-dimensional models are particularly important in structural geology where geometry is a major observable factor and steep slopes carry much of the 3D geometric information. Terrain model visualizations are widely used today (e.g., Google Earth and in Geographic Information System (GIS) software), yet most of these visualizations are not true 3D, and in steep terrain, can contain serious errors that may lead to misinterpretations, e.g., [2,3]. Light Detection and Ranging (LiDAR) systems address many of these problems with high-resolution terrain models, yet LiDAR data are expensive to acquire and are flawed when applied to most bedrock geology studies [2]. All the digital 3D visualization methods provide information on an area that can be preserved indefinitely and add the ability to digitally revisit field sites to document changes or review previous observations with newer digital tools to enhance the 3D model. Note that these types of visualizations ultimately should advance the efficiency of field research by minimizing the need to revisit sites several times because critical information may not be sufficient for newer scientific interests. Thus, the need to repeat surveys or re-map areas with ground-based methods to collect some new information is rapidly being replaced by a variety of digital mapping techniques that can now provide 3D visualizations at some level of accuracy.

One of the more recent developments is the use of a modern photogrammetry method to build 3D terrain models from a suite of overlapping images that capture views from a variety of look directions. The images are merged to generate a quantitative, 3D representation of Earth's surface, regardless of terrain slope [4]. The method is commonly referred to as Structure-from-Motion, but this term misrepresents the process. In reality, the method is based on two distinct processes [5]: (1) a feature-matching algorithm that recognizes common features on a suite of photographs, then uses optical theory to calculate positions of key points and camera parameters as a first step (the Structure-from-Motion step) and (2) a model refinement step (Multiview stereo step) that fills in the model between key points with additional photo data to produce a high-resolution model. The Multiview stereo step is particularly important because it produces a dense, photo-realistic point cloud that can be directly viewed in a 3D visualization. We refer to the entire process as SM (Structure-from-Motion/Multiview stereo). SM has become an important tool for geology because it provides realistic 3D visualizations at low cost with no requirement for special, expensive equipment. Photographs can be obtained relatively easily from a variety of sources including handheld cameras, drones, aircraft, or satellites. Geologic applications range from detailed imaging of small fossil beds, e.g., [6], to imaging of large areas, e.g., [2,4,7–10]. Moreover, the ease with which photos can be collected and processed into 3D models makes SM potentially transformative for field geology applications by affording unprecedented resolution of geologic structure, e.g., [2,8–12]. Nonetheless, when and where 3D photo-based models can be used and how they compare with LiDAR-based models for geologic field studies remains uncertain because comparative studies, e.g., [13],

are relatively rare and geologists are just beginning to explore the full potential of some of the data acquisition techniques.

Most studies of 3D model accuracy have focused on engineering applications or high-resolution topographic analyses in geomorphology, typically using high-resolution ground control to constrain the model, e.g., [14–18]. This level of accuracy, however, may not be necessary or practical for many geologic field studies where logistical considerations limit access. Nonetheless, measurements of orientations and accurate mapping of the geometry of geologic bodies are a critical part of field geology. Thus, it is important for the geologist using a 3D model to know that the model is scaled properly and free of distortion as well as oriented properly for digital measurements to be directly compared to those made in the field.

We and our students, e.g., [2,3,6,8,19], have spent much of the last decade experimenting with the collection and processing of SM data from a variety of geologic settings, and we now have a number of sites where multiple methods have been applied. We use some of these data here to establish some preliminary information on the accuracy and resolution of 3D models based on different data acquisition methods. This is intended as a starting point for determining what methods work for solving a variety of problems.

In this paper, we analyze the accuracy and resolution of SM point clouds emphasizing a data set from the eastern California desert (Figure 1). These data include a LiDAR survey acquired with a Terrestrial Laser Scanner (TLS) that was acquired early in the project through assistance from UNAVCO. The LiDAR data serve as a basis for comparison to SM data acquired from ground-based imaging, drone-based imaging, and imaging acquired using a chartered fixed wing aircraft. Merged data sets are also examined for multiple drone flights merged into a single model. Our emphasis is on techniques that do not require any ground control placement because that process can be time-consuming and potentially dangerous or logistically impossible in remote field settings. Thus, our intent is not an analysis of high-resolution surveying methods; rather, our emphasis is on determining the accuracy of SM surveys acquired ad hoc, using only conventional GNSS systems and access to applications such as Google Earth or GIS for georeferencing. We begin with a description of our data acquisition and data processing methods and then assess the different methods. We then compare the models quantitatively and assess sources of error. We end with a discussion of the pros and cons of different methods in the context of field geology.

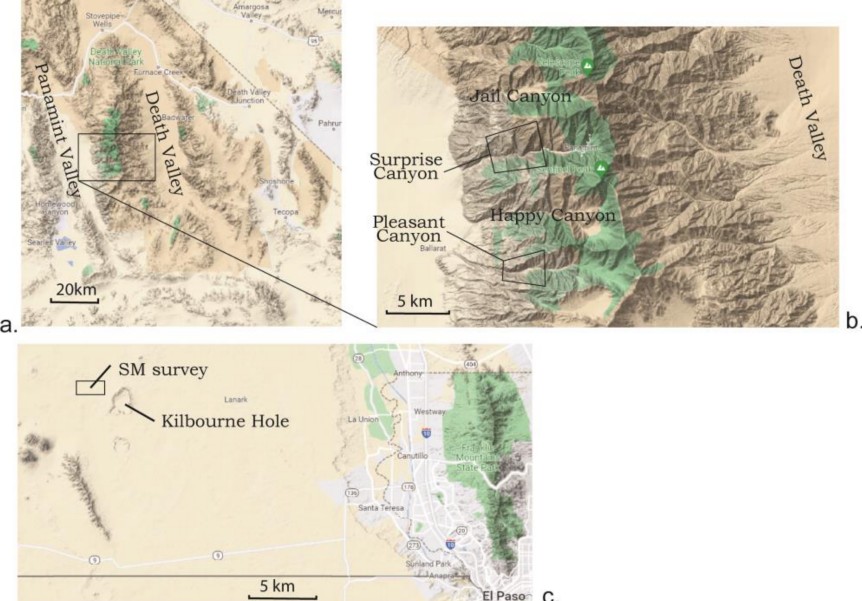

**Figure 1.** Location maps for the study. (**a**) shows location of the surveys within the Death Valley region, CA, USA, with (**b**) showing detailed locations of the canyons where surveys were conducted. (**c**) shows location of the reference survey in southern New Mexico, west of El Paso, TX, USA. Base map is from Google Maps.

## 2. Background and Methods

### 2.1. Background

Stereo aerial photography was the first major tool in photogrammetry when camera optics were quantified and used as a basic tool to produce regional topographic maps [20]. By the late 20th century, computational power had progressed to allow the development of terrain models and orthoimagery using vertical incidence photogrammetric methods, e.g., [20]. At nearly the same time, radar techniques were applied to low earth orbit missions, most notably the space shuttle missions of the 1990s, to produce digital elevation models (DEMs) of most of Earth's surface (https://www2.jpl.nasa.gov/srtm/, accessed on 29 May 2023). Together, these technologies afforded the first 3D visualizations of Earth's surface at a global scale, including the introduction of online viewers such as NASA's WorldWind site and Google Earth. These tools were important for the geosciences by affording a 3D view of virtually anywhere on the planet. Nonetheless, these tools had limited application for geologic mapping because they were typically of low resolution.

GIS software developed during this time interval also provided powerful tools for geologic mapping, e.g., [1], yet these tools remained flat-map centric. A widely used exception to the flat-map approach arrived in 1999 when ESRI released ArcGIS, which included Arcscene, a 3D viewer based on draping orthoimagery onto a DEM. This approach to the 3D visualization of Earth's surface is not true 3D however, and is often referred to as a 2.5D method [2] or pseudo-3D. Specifically, an orthocorrected image with parallax removed to make a flat-map representation of the image is projected vertically onto an elevation model to produce the pseudo-3D view. It is pseudo-3D because the accuracy of the projection is dependent on the accuracy and resolution of the terrain model as well as the orthocorrection of the imagery. Thus, in areas of steep terrain, this method often produces artifacts such as pixel smear and scenes that do not resemble reality, e.g., see examples in [2], and off-nadir images can introduce artifacts that look real but are highly distorted [3].

LiDAR is the first technology that was developed to accurately locate points on the Earth's surface with high accuracy because it is based on measurements of the travel time of a light wave between a scanner and the surface being mapped. It can be acquired from either an airborne platform or a terrestrial scanner (TLS) with the primary source of error coming from the accuracy of the scanners known location for each measurement and the distance between the scanner and the point of measurement on a surface. In geosciences, the main applications of high-resolution LiDAR terrain models have been in geomorphology, measuring surface morphology and surface changes through time based on repeat surveys, e.g., [21,22]. Although LiDAR has been very important for geomorphic studies, it has seen less use in bedrock geologic studies because it is often expensive and requires either heavy field equipment in terrestrial surveys or an airborne platform. In addition, LiDAR was not designed to acquire true color visualizations of the measured surfaces and, as a result, photographic images need to be added to a LiDAR model by draping them onto the surface model. Thus, as with DEM-based methods, the resultant model is pseudo-3D when an image is draped on the model. The high accuracy of the LiDAR surface models often reduces the need for very closely spaced measurements, and 3D scene visualizations often use a mesh surface to fill in the spaces between measurements and then imagery is draped onto the mesh. Although these methods can produce stunning visualizations, they produce challenges for studies such as bedrock geology, where the quality of the imaging is critical and minor variations in features can be significant in terms of measuring and observing small-scale features.

SM technology solves the visualization limitations of LiDAR that handicap bedrock studies because the data acquisition can be tuned to ideal photographic conditions rather than things such as an arbitrary time schedule and aircraft flight limitations. Moreover, the only equipment required is a camera, although airborne platforms and GNSS positioning aid data acquisition and processing to greatly improve SM model creation. That is, the variations in look directions and proximity to the surface that can be provided by using

aerial imaging and the use of a GNSS system to georeference the camera position can significantly improve processing and model accuracy. Data acquisition and the processing of SM data have been extensively reviewed elsewhere, e.g., [3,14]. For our purposes here, the key feature of SM data is that when a point cloud is generated in an SM survey, every point in the cloud will have its true color (typically an RGB value) assigned to it, such that each point can be thought of as a 3D pixel on an irregular surface [2]. This feature is critical for applications to bedrock geology because a user of the data can be confident the data are free of artifacts from improper image drapes, e.g., a color change seen within a family of points in a point cloud is in its proper position in space, free of distortions from image drape. Virtually all of the operations that a field observer makes, except the collection of samples, can be made using the SM model if it is reasonably accurate.

Unfortunately, SM technology is sufficiently new that less is known about the accuracy of these data relative to LiDAR [4]. There is significant information on the subject in the engineering literature, e.g., [14–18], yet in many respects, these studies are not applicable to geological mapping applications outside of geomorphology. For example, most engineering studies have focused on the absolute accuracy of SM models, particularly the accuracy of the georeference, e.g., [15,17], and virtually all use high-resolution ground control points for model referencing. The placement of ground control points is often difficult or impossible for a geologic mapping project in rugged terrain. More important for geologic studies is the accuracy of relative positions such that a 3D model is a true representation of geometry free of distortions (straight lines are not curved) and scaling errors. Wilkinson et al. [23] provided an assessment of some of these issues in virtual outcrop models for geology and found that properly acquired SM data typically lacked scale distortions but did suffer in areas where topographic features produced sharp corners. Our experience at virtual outcrop and larger scales is consistent with that conclusion and arises because the feature-matching algorithms in the SM software have difficulty tracking shape changes in objects that accompany changes in look direction [4]. Here we emphasize that absolute positioning errors are most likely to be important in SM studies at larger scales than the virtual outcrop level. That is, in geologic studies where adjoining, independent surveys are merged, absolute errors could cause a mismatch that would need to be reconciled to make a larger, 3D model. Note, however, that these kinds of absolute errors are likely small when data are constrained by modern GNSS technology. Thus, the questions we consider here are: how large are absolute errors and relative errors in SM data acquired under different acquisition methods? How do errors vary under different georeferencing techniques, and across scale ranges? What level of accuracy is "good enough" for a typical geologic problem with today's technology? We address these questions here with direct comparisons between SM data and a TLS survey. In addition, a critical concern for field geologists is what can be seen and measured (resolution) in an SM model compared to true field observations and what is needed in an SM survey to achieve a desired resolution.

### 2.2. Methods

The data for this study consist of a variety of surveys conducted over a period of approximately 10 years in the Panamint Mountains of southern California (Figure 1) to support geologic mapping in an area of rugged terrain. The geologic structures of interest are complex and are well exposed on steep slopes and cliffs that are extremely difficult to access. This is the ideal setting to apply remote sensing techniques that are highly reliable.

The initial work involved collecting ground-based LiDAR data, geologic mapping, and ground-based SM acquisition and interpretation, e.g., [3]. This was followed in subsequent years by flying drones, first in Surprise Canyon where geologic mapping had previously been performed but many of the more remote areas had not been accessible for direct observations. The drone study was expanded into Pleasant Canyon and more data were collected in Surprise Canyon in 2021 and 2022 with newer models of drones with higher quality cameras and onboard georeferencing. An autonomous flight was used with one of the new drones in Pleasant Canyon. Finally, in 2021 we used a low-flying aircraft with

handheld cameras to collect additional data over Surprise, Jail, Happy, Hall, and Pleasant Canyons (Figure 1). These various surveys did not all overlap, and there are variations in lighting conditions that affect the comparison of these data sets.

As a comparison to these data collected in steep terrain, we also analyzed a data set from southern New Mexico (Figure 1), west of El Paso, TX. This area was recently covered in a small SM survey within an area where the USGS recently acquired airborne LiDAR data. The New Mexico site has relatively low topographic relief so it shows how LiDAR and SM models compare in a more typical setting.

### 2.2.1. LiDAR Data

In this study, we analyze the accuracy of a series of SM surveys relative to LiDAR data under the assumption that the LiDAR data are a high-resolution reference that accurately maps positions (reflecting points) on a surface at the cm level. This accuracy assumption is based on the accuracy of GNSS used to locate the scanner and reference the survey point positions relative to the scanner. Our primary reference survey is a TLS data set from the Panamint Mountains (Figure 1) that was acquired in 2013 as part of experiments in 3D geologic mapping. These data are available at opentopography.org. A description of data acquisition and processing is in Brush et al. [3]. For this paper, the key information for these LiDAR data are that the scanner positions were accurate to 2–3 cm with point positions accurate to ~1 cm relative to the scanner, indicating the LiDAR point cloud is accurate to 3–4 cm overall [3].

Because this reference survey is TLS data, scanner positions were controlled by logistics. In this case, the survey sites were all located on the floors of canyons, each with slightly different characteristics: (1) Pleasant Canyon is a steep-walled canyon ~300–600 m deep and 800–3000 m across that is accessible by a road on the canyon floor; (2) lower Surprise Canyon is only accessible on foot along a trail at the bottom of a deep, steep-walled canyon with a tight inner canyon (~200–700 m deep and ~250–300 m across) and a broader canyon overall (total depth of ~500–1300 m and ~3000–4000 m across).

### 2.2.2. SM Data

For SM, our principal data are a series of photographic site surveys conducted between 2013 and 2022 that overlapped with the TLS survey area. Our initial work was strictly ground-based photography with details in Brush et al. [3]. More recently, we acquired two aerial SM data sets in this area: (1) a group of drone-based surveys using a DJI Mavic 2 Pro drone as well as a DJI Mini 2 drone and (2) an aerial survey using a chartered, fixed-wing aircraft with imagery acquired using handheld cameras imaging through open windows on both sides of the aircraft.

Drone Imaging: For the drone imagery, we acquired data using different methods as experiments on best approaches to use in a deep canyon with steep walls. We used two drones (DJI Mavic 2 Pro and DJI Mini 2). The Mavic Pro's lens is a 28 mm equivalent of a 35 mm film camera and a still image size of 5472 × 3648 (https://dominiondrones.com/pages/dji-mavic-2-pro-specifications, accessed on 29 May 2023). The Mini 2 has a lens that is a 24 mm equivalent of a 35 mm film camera with an image size of 4000 × 3000 pixels (https://www.dji.com/mini-2/specs, accessed on 29 May 2023). Because of ease of access and relatively "open sky" conditions where GNSS operates best, our main experiments were in Pleasant Canyon, but we also conducted experimentation in Surprise Canyon.

Most of our drone flights were launched from the valley floor for the same reason the TLS data were acquired from the valley floor. In tight canyons such as these, the canyon geometry was a challenge for line-of-sight flights, and we developed a method where the flight begins with a low-elevation flight paralleling the cliff face, while the drone camera is pointed at the cliff. In this pass, the drone could easily be kept in sight either directly or by walking with the drone. Once this pass is completed, continuing the flight was aided by a feature of the DJI flight software that provides a map view showing the track of the drone. This track allows a procedure where the track can be used as a reference as the

drone is flown to successively higher elevations, by following a comparable track with each pass, typically moving toward the cliff face with each higher pass and using live video to evaluate safety of the closer approach. The result is a semi-vertical grid pattern to the flight with imagery acquired in successive sub-horizontal passes that parallel the cliff face at similar distances. This pattern mimics autonomous flight patterns used on flat terrain in programs such as drone deploy (www.dronedeploy.com accessed on 29 May 2023) or in Agisoft Metashape flight plans for imaging vertical features (www.agisoft.com accessed on 29 May 2023). This imaging pattern avoided problems we recognized in ground-based imaging [3] and in poorly executed drone flights (see below).

Within this series of experiments on imaging geometry, we also conducted experiments on how the actual images are acquired. The simplest method is to set the drone to interval shooting, which, on both our drones, is a minimum of 2 s, and that value was used in all our experiments. Although this method is simple and assures continuous coverage, it can lead to numerous redundant images (e.g., when the drone is stopped in flight) and potentially image blur if the drone is flown too fast at close range. Thus, we experimented with other approaches including: (1) a manual equivalent of interval shooting where the drone is flown short increments, stops, and an image is acquired manually, and (2) a modification of this manual shooting where at each stop position, the drone is rotated on a vertical axis using the controller and horizontal axis via the camera gimble. The second method is an aerial equivalent of a method we used in ground-based images with a relatively long lens where scenes are panned from individual camera positions (2). Ultimately, we found the first manual method a waste of flight time because we used high enough shutter speeds that image blurring was minimal to nonexistent. Based on our ground-based methods, we expected the second method to produce superior results but ultimately saw no significant difference from simple interval shooting. A more controlled experiment shooting the same scene by the two methods might reveal distinctions, but our qualitative assessments suggest the increased efforts for the pilot make manual imaging methods less desirable.

Finally, at one site, we were able to conduct an autonomous flight experiment by launching the drone from the canyon rim. In this flight, we utilized the program "drone deploy" and flew a constant elevation survey at a height of 100 m above the starting point. The survey imaged an area outside the coverage of our TLS data but overlapped extensively on a cliff face, allowing assessment of a vertical imaging survey compared to the TLS data acquired from the canyon floor.

Imaging from a chartered aircraft: In addition to drone imagery, we conducted a low-elevation (~400 m to 800 m above terrain) imaging flight in March 2021 with a chartered fixed-wing aircraft (Cessna 180). In this flight, two people, each looking out opposite sides of the aircraft, acquired oblique images through open windows on the aircraft with a camera. This flight produced a serendipitous experiment when both GPS units attached to the cameras failed to provide continuous GPS positioning, leading to a flight with unreferenced camera positions. The cameras were Nikon DSLRs: a D5300 and a Df. Both cameras had Nikkor fixed focal length lenses that approximated a 50 mm lens in a 35 mm film camera. The Nikon Df contains a full frame sensor, and thus, its 50 mm lens was 50 mm senso stricto, whereas the D5300 has a smaller sensor and we used a 35 mm lens to approximate a 50 mm lens. Both cameras were operated on a fixed f-stop with exposures adjusted by the camera through variable shutter speeds. The cameras were set to record in both jpg and raw modes, with the latter providing improved image processing. The raw format proved particularly important when data from the two cameras were merged due to differences in the sensors producing different appearances of the images, which could be easily adjusted in the raw imagery. During image acquisition, each operator collected imagery using a vertical sweeping motion that ranged from 20–70 degrees, depending on the flight position relative to the scene. The flights were flown in a grid pattern with 3 passes parallel to the mountain front and 4 cross-lines flown down four major canyons at a height about midway between the canyon rim and the canyon floor. Note that because of limitation of the aircraft climb rate, imagery was only acquired in downcanyon flights.

Drone control experiment, Kilbourne Hole, New Mexico: As a comparison to the models obtained in the steep terrain of the Panamint Mountains, we flew a small drone survey over the low-relief volcanic field northwest of Kilbourne Hole in southwestern New Mexico (Figure 1). The area was chosen because it had been recently covered in an airborne LiDAR survey (https://www.usgs.gov/3d-elevation-program, accessed on 29 May 2023), allowing direct correlation to an SM survey over the same area. The survey was flown at a height of 100 m using the Mavic 2 pro drone flying autonomously using drone deploy software.

### 2.2.3. Data Processing

In this project, we processed the SM data using both Metashape 1.8 and Pix4D mapper 4.6.4 software. We experimented with a variety of data processing techniques. Aside from point density variation under different options in the Multiview stereo step (described below), the variations in processing parameters had little effect on the model accuracy in almost all cases. One exception was recognized in Metashape when the parameter "camera accuracy" was varied in the Structure-from-Motion step. Specifically, when camera accuracy is set relatively low (10 m or more), the "alignment" of images can improve, but model accuracy can degrade and produce distortions on drone models. When set to 2 m or less, however, these problems disappear.

Because our GNSS system failed in our manned aircraft data, these data were not georeferenced by camera position, which afforded a test of a likely scenario when SM data are acquired ad hoc, or ground control placement is impossible. For example, we have worked in remote settings accessible exclusively by aviation, and in these settings, ground control placement is not possible, and ad hoc imaging might be performed literally "on the fly" when some feature is observed out the window of an aircraft. Thus, the method used here is potentially relevant to a range of future SM data acquisitions.

In both Metashape and Pix4D, we used a similar georeferencing procedure with these aircraft data. We began by running the SfM and MvS steps with unreferenced images to build a low-density point cloud. We then compared our model to Google Earth visualizations of the same area and identified 10–15 potential ground control points (GCPs) visible in both scenes. In these scenes, we used exclusively natural objects, but other GCPs are possible where man-made structures are visible (e.g., buildings or road intersections). For GCPs, we focused on objects 1–3 m in size or larger objects that had recognizable shape patterns where a common point could be easily located. Google Earth provides coordinates by placing a marker at the object, which can then be entered manually into SM software when the comparable point is picked on the SM model. Both Pix4D and Metashape have features where once the GCP is picked, individual images can be queried for quality control on the point placement. We experimented with different GCP placement schemes and processing streams that might arise under different situations and, not surprisingly, a range of referencing issues arose.

It is of note that Google Earth uses an unusual georeference that made us suspicious of using Google Earth for referencing. Specifically, Google Earth uses a pseudo Mercator projection with a WGS84 datum for horizontal reference and variable DEM resolution with a geoid reference (https://en.wikipedia.org/wiki/Web_Mercator_projection, accessed on 29 May 2023), although which geoid is used is not clear in documentation we have seen. Because of the uncertainties in the datum, particularly vertical where a low-resolution DEM could introduce large errors, we were uncertain on how well this referencing technique would work. Nonetheless, because this is likely the simplest procedure for field geologic studies, we used this method exclusively with the results reported here.

### 2.2.4. Point Densities and Cloud Mismatch Measurements

For comparison of results among the different acquisition and processing methods we used the open-source program, *CloudCompare*, to analyze point clouds for point density and distance between SM point clouds relative to the TLS data. We also used Maptek

Ltd.'s program, *PointStudio*, to compare point clouds locally as a visual check on the CloudCompare results and used PointStudio for fine cloud alignments where point clouds were merged to form a single model outside of SM processing software.

## 3. Results

### 3.1. Point Densities

Animation 1

Point density is the number of data points within a specific area or volume of a model divided by the area or volume. It may vary throughout the model and provides an estimate of the resolution of the area of the model because the more points within a given area, the more likely it is that one or more points will image a small feature. The model is unlikely to image objects smaller than twice the inverse of the point density and spatial aliasing is possible at these resolutions. In our models, the point density was not specifically considered during the data collection, but we knew that the distance between the camera and the surface was inversely proportional to the resolution and that camera parameters would affect resolution. By simple visualizations of our models (Figure 2 and https://www.youtube.com/watch?v=Ajk34HKzGDo, accessed on 29 May 2023), it is clear that point density varies markedly among the models, depending on the method.

## Effects of Variations Point Cloud Density on Visualization

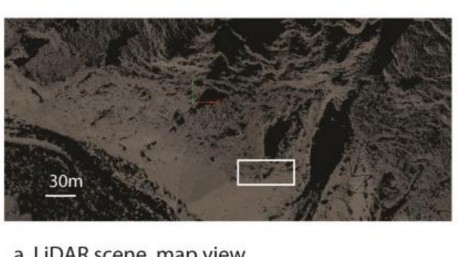

a. LiDAR scene, map view

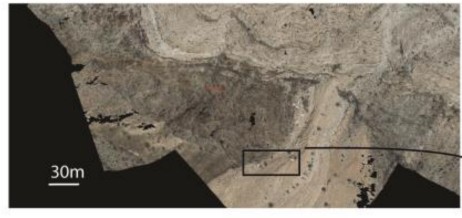

b. same view as a., single flight Mavic Pro SM model

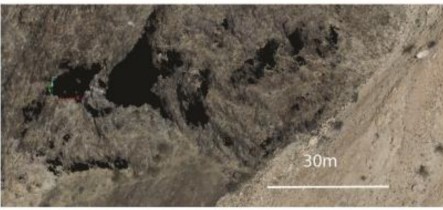

c. same flight as b., Mavic Pro SM model, oblique close view

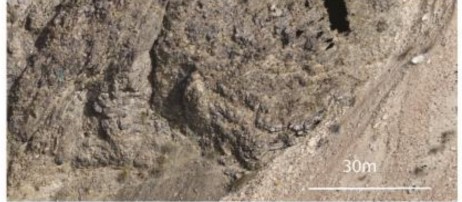

d. same scene as c., Mini 2 SM model

**Figure 2.** Illustration of effects of variations in point density on ability to resolve geologic features in 3D mapping onto a point cloud. Scenes are from lower Pleasant Canyon. (**a**,**b**) are a map view visualization of the same area showing the poor visualization in the relatively low density of the LiDAR scene (**a**) vs. SM model (**b**) developed from a single Mavic Pro flight. (**c**,**d**) show a zoomed in, oblique view of a part of the scene shown in (**a**,**b**) (box shows location). (**c**) shows the zoomed in Mavic Pro SM model and (**d**) shows the zoomed in image from the DJI Mini 2 model. There is no image of the LiDAR zoomed in because it is too sparse to show at this scale. Similarly, the DJI Mini 2 model is not shown at the smaller scale because it looks the same as the Mavic Pro at that scale but differences are apparent at the larger, zoomed in scale. Point densities are quantified in Figures 3 and 4 for these data.

### Pleasant Canyon LiDAR Point Densities

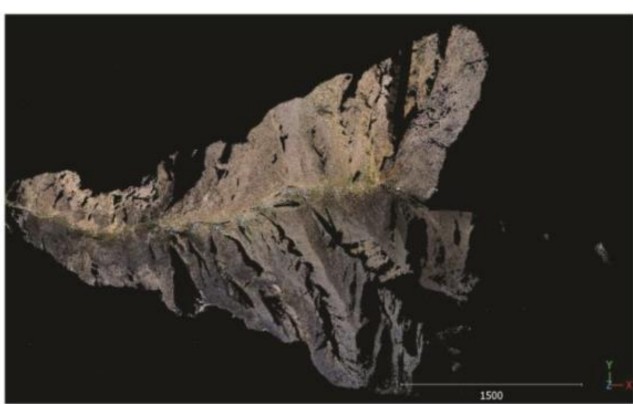

a. Raw Point Cloud (RGB from camera on scanner)

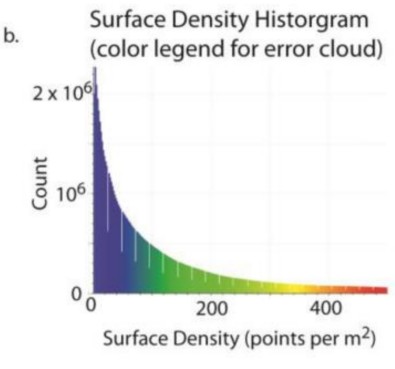

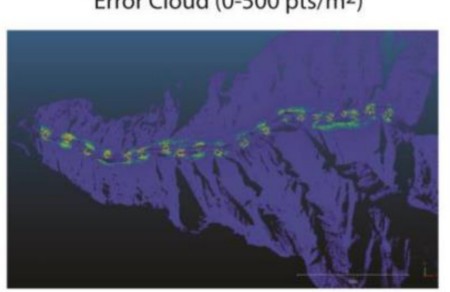

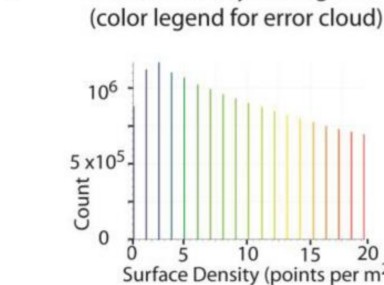

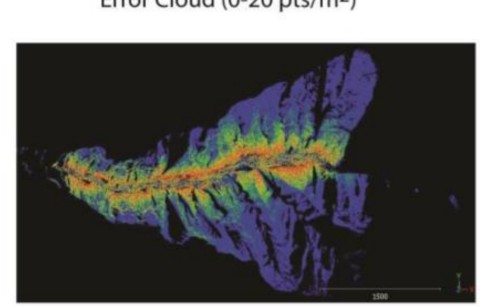

**Figure 3.** Point density analysis for LiDAR data in Pleasant Canyon illustrating decrease in point density with distance in TLS data such as these. All views are map view of 3D data. (**a**) shows the raw point cloud colored with RGB values obtained from a camera on the scanner, (**b**) shows error cloud (**right**) colored by the surface density histogram (**left**). Note that high-density areas show locations of scanner positions. (**c**) splits the point cloud into low point densities recolored to histogram (**left**) in the error cloud (**right**).

Fleming and Pavlis [19] and Brush et al. [3] discussed the pros and cons of high-density point clouds vs. textured mesh models as a base for 3D geologic mapping. These studies both emphasized the advantages of high-density point clouds because areas of sparse data are more obvious and afford visual assessments of mapping accuracy not possible with mesh models where the image drape can produce a false sense of high accuracy, even in areas where the data are sparse. The animations of our Surprise Canyon data (https://www.youtube.com/watch?v=Ajk34HKzGDo, accessed on 29 May 2023) and Pleasant Canyon (Figure 2) show, however, that the viability of a point cloud for visualization depends on point density. Software can partially compensate for this limitation by enlarging the point size in the visualization, but that visualization option has limited capability to compensate for sparse data. Thus, because point density will strongly effect the ability to

interpret the 3D terrain model, we analyzed point density among the different methods (Figures 2–6 and Table 1).

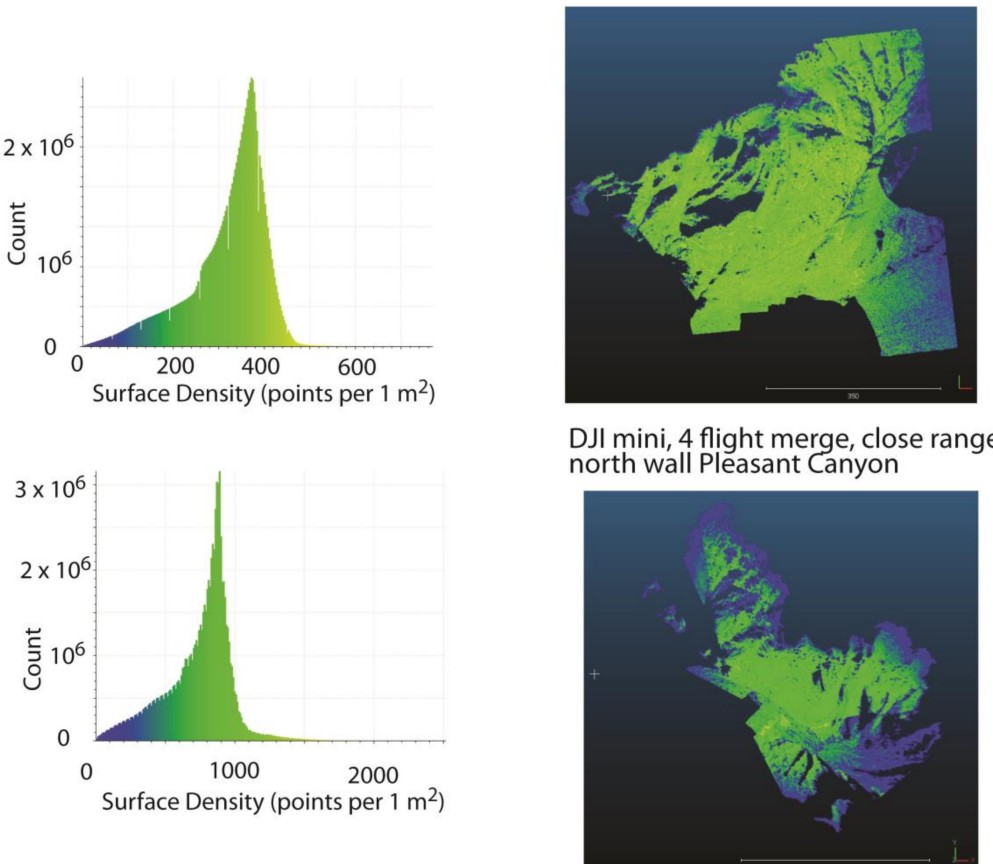

**Figure 4.** Quantification of point densities per square meter for two drone-based models in Pleasant Canyon that were shown qualitatively in Figure 2. Both drone models were processed to high-density point option in Metashape and displayed here in map view. Upper figure illustrates typical point densities for the Mavic drone at intermediate distances (100–200 m) in a single flight vs. a close range (10s of m) model for the mini merging 4 flights. Note the two figures are approximately at the same scale but overlap only about 30% along the eastern part of the DJI mini model and western part of Mavic model as in Figure 2.

Note that point density is a crude measure of point spacing and, therefore, model resolution, e.g., 100 points/m$^2$ is ~10 cm point spacing, whereas 400 points/m$^2$ would be ~5 cm point spacing, etc. CloudCompare provides three ways to measure point density: surface density, volume density, and measured neighbor density (https://www.cloudcompare.org/doc/wiki/index.php/Density, accessed on 29 May 2023). We primarily used surface density for this study, which is a measure of the number of points within a given area. In this method, the program places a measurement circle centered on each point and oriented by neighboring points and counts the number of points falling within the circle. In our case, we used a 0.56 m radius (1 m$^2$) circle for high-resolution models and a 2 m radius circle for lower resolution models. We tested the surface density vs. neighbor density on one site with a high-density point cloud and found no significant difference between the two methods. The neighbor density was useful, however, in assessing point densities of our LiDAR point clouds that fall off in density with distance from the scanner. The results of this analysis are displayed in Figures 2–5 as a colored point cloud with an accompanying histogram of density vs. count with a color ramp in this histogram corresponding to the colored point cloud. Table 1 summarizes the results from all our data analysis.

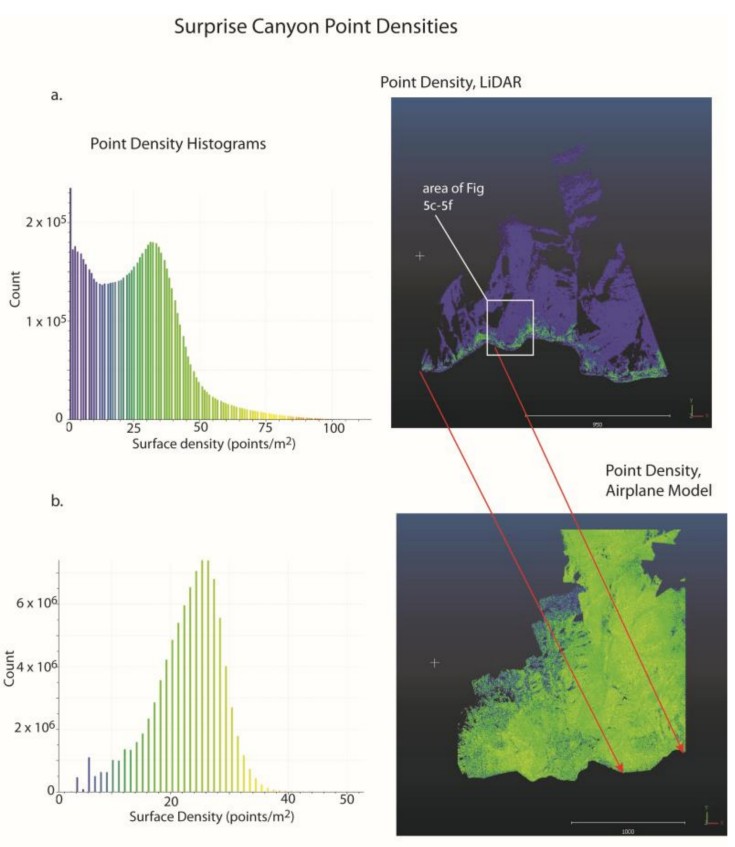

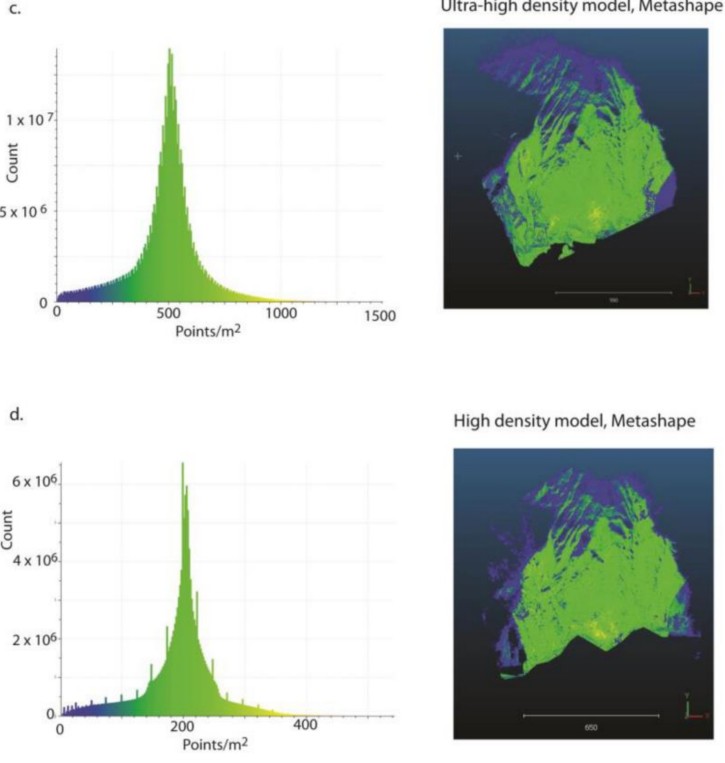

**Figure 5.** *Cont.*

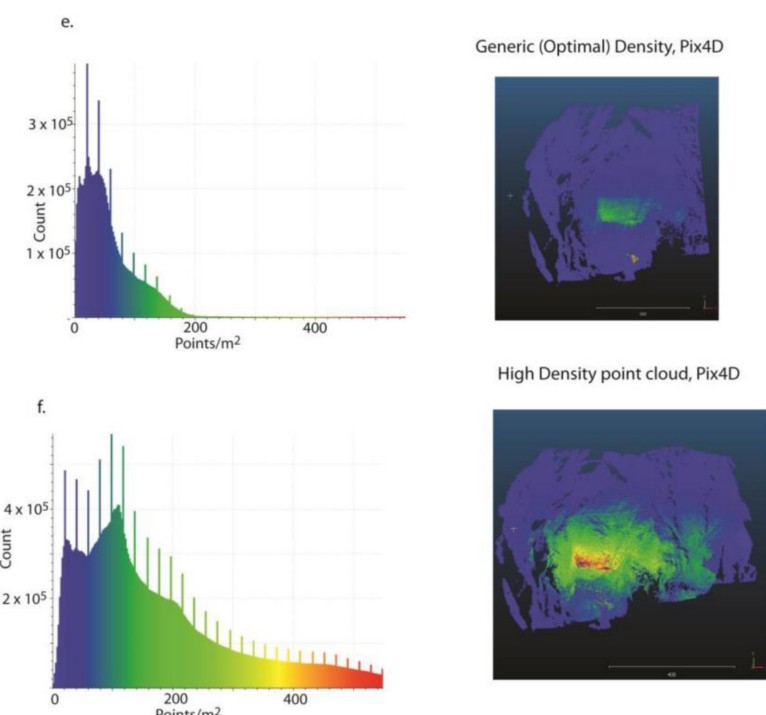

**Figure 5.** Point densities for the part of the Surprise Canyon data analyzed in this study. All figures show surface density as a color map on the right with a histogram of point density on the left indicating the color ramp for the figure to the right. (**a**) is for the TLS data and (**b**) is for the SM data acquired in the airplane flight. Red lines show approximate equivalent points on the TLS and SM airplane model. (**c**,**d**) show point density comparisons for an SM model processed using the utra-high point density option (**c**) and high-density option (**d**) in Metashape software. Scene is from a single flight with the Mavic Pro drone and a relatively small number of images (178). Ultra-high-density scene is visualized in https://www.youtube.com/watch?v=Ajk34HKzGDo (accessed on 29 May 2023). (**e**,**f**) show point densities for the same scene as 4c and 4d but processed in the two point density modes in Pix4D software.

The TLS data were clearly the sparsest (Figure 2 and https://www.youtube.com/watch?v=Ajk34HKzGDo accessed on 29 May 2023) and Figure 2 illustrates this quantitatively. Note that the relatively low density of the TLS data and presence of large holes in the data is a function of two issues that are related to the logistics of data acquisition: (1) the data were acquired on a canyon floor, which limited each scan to a $1/r^2$ point spacing, decreasing the point density dramatically with distance from the scanner sites as well as limiting look direction; and (2) time/cost restrictions led to a compromise on the spacing of scanner positions. The decrease in point density with distance is clearly shown in Figure 3 where all of the high point densities are within a few meters of the scanner (Figure 3b) and even the close in data (Figure 3c) on the cliff face are in the 5–20 points/m$^2$ range with the upper slopes all at low densities of <5 points/m$^2$. The Surprise Canyon LiDAR data are comparable (Figure 3 and Table 1). This point density is as expected from the data acquisition because LiDAR data are limited by the geometry of the scanner position vs. distance with no way to fill information between measured points other than increasing scan density with accompanying increased data collection time. In contrast, the Multiview stereo step in SM data can fill in voids, as long as there are enough look directions to cover a 3D scene. This distinction is shown in the analysis of SM data (Table 1 and Figures 3–5).

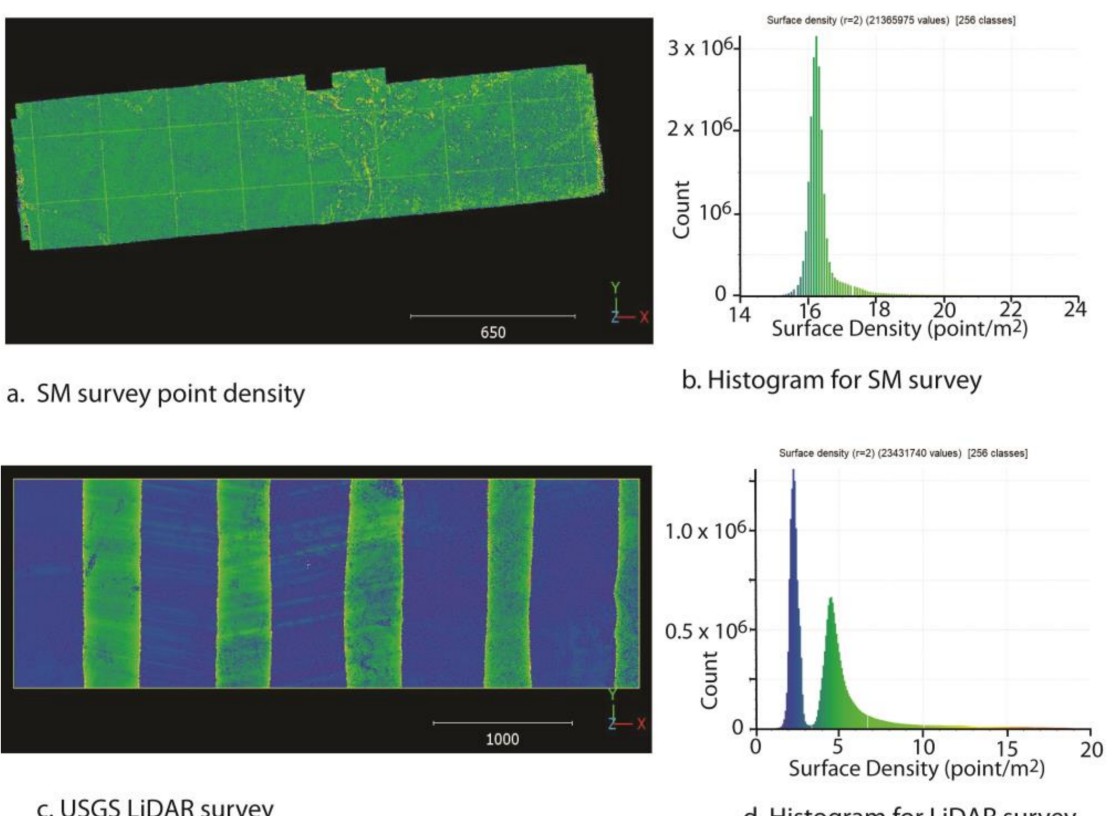

**Figure 6.** Point density analysis of an SM model developed from a vertical imaging, autonomous drone flight flown at 100 m vs. USGS LiDAR survey (www.usgs.gov/3d-elevation-program, accessed on 29 May 2023) flown over the same area. Drone flight covers a subset of the LiDAR coverage. (**a**) shows a map view point density distribution in SM model based on color scheme in histogram of point densities shown in (**b**). Note the average point density of the survey of ~16 points/m$^2$. (**c**) shows a map view point density distribution for the LiDAR data colored based on the color scheme of the point density histogram (**d**). Note the stripped pattern reflects NS flight lines with general point densities (blue) ~2.5 points/m$^2$ and two times higher densities (~5 points/m$^2$) in areas of data overlap (green strips).

In Figure 4, SM models from two different drones in Pleasant Canyon but with different flight plans are illustrated: a close in (10s of m from target) DJI mini composite model and a single flight with the Mavic Pro at intermediate distances (100–200 m from target). These data confirm our observation from visual inspection that our LiDAR data (https://www.youtube.com/watch?v=Ajk34HKzGDo accessed on 29 May 2023 and Figure 2), in general, are approximately an order of magnitude lower density than any of the drone data. That is, the LiDAR point densities are less than 20 pts/m$^2$ (point spacing > 20 cm) and <5 pts/m$^2$ for more than half the model, whereas the drone data range from 300–1000 pts/m$^2$ (point spacing ~3–6 cm). The distinction in the close in DJI mini vs. Mavic flight (Figure 2) also illustrates the effects of range and number of images used in model construction (487 images for mini flown at ~50 m from the target vs. 186 images for the Mavic flown at ~200 m from the target).

**Table 1.** Point densities and point cloud spacing for different methods.

| Location | Acquisition Method | Point Density Range | Point Density at Peak Count | Estimated Resolution at Peak Count |
|---|---|---|---|---|
| | **LiDAR** | **(Points/m$^2$)** | **(Points/m$^2$)** | **cm** |
| Pleasant Canyon Brush et al., (2018) | TLS | 1–100 at target (spatial dependence) | <20 | >22 |
| Surprise Canyon Brush et al., (2018) | TLS | 1–100 at target (spatial dependence) | <20 | >22 |
| Kilbourne Hole, NM See USGS metadata | USGS airborne LiDAR, low-relief terrain | 2.5 (background) to 5 (overlap areas) | 2.5 | 63 |
| | **Aircraft** | | | |
| Pleasant Canyon-HD Metashape | Manned aircraft, Nikon Df camera, single pass | 1–15 | 10 | 32 |
| Surprise Canyon HD Metashape | Manned aircraft, two cameras | 10–40 | 25 | 20 |
| | **Handheld Camera** | | | |
| Surprise Canyon HD Metashape (Brush et al., 2018) | Ground-based, Nikon D5300 at 200–700 m | 1–500 | 290 | 6 |
| Pleasant Canyon HD Metashape (Brush et al., 2018) | Ground-based, Nikon D5300 at 600–1500 m | 1–80 | 45 | 15 |
| | **Autonomous Drone flt** | | | |
| Kilbourne Hole, NM HD Metashape | DJI Mavic autonomous flight 100 m elevation | 15–18 | 16.2 | 25 |
| | **UAV (Drone)** | | | |
| Pleasant Canyon HD Metashape | DJI Mavic, 100–200 m from target, single flt. | 1–400 | 375 | 5 |
| Pleasant Canyon HD Metashape | DJI mini, close range (10–30 m) | 1–1200 | 850 | 3 |
| Pleasant Canyon HD Metashape | DJI Mavic, 4 flight merge | 1–800 | dual max at ~210 and 390 | 7 and 5 |
| Surprise Canyon HD Metashape | DJI Mavic, single flight | 1–400 | 200 | 7 |
| Surprise Canyon Ultra HD Metashape | DJI Mavic, single flight (same as above) | 1–1000 | 500 | 4 |
| Surprise Canyon Pix4D (optimal) processing | DJI Mavic, single flight (same as above) | 1–200 | 20 | 22 |
| Surprise Canyon HD Pix4D | DJI Mavic, single flight (same as above) | 1–600 | 100 | 10 |
| Surprise Canyon HD Metashape | DJI Mavic, two-flight merge in canyon bend | 1–230 | 170 | 8 |

In Figure 5, we illustrate the model resolution of the LiDAR, SM model from the airplane flight, and drone data under different data processing schemes using data from Surprise Canyon (area shown in https://www.youtube.com/watch?v=Ajk34HKzGDo accessed on 29 May 2023). Figure 5a confirms the observations for the LiDAR resolution in Pleasant Canyon (Figure 4) with similar model resolution, including a higher resolution close to the scanner near the canyon floor, but point densities <10 points/m$^2$ (point spacing >~30 cm) anywhere more than a few 10s of m from the scanner. The airplane model

(Figure 5b) demonstrates that this model overlaps in point density with the LiDAR model but has a more uniform distribution of point density with a histogram peak that is 3 to 4 times higher than the point density in most of the LiDAR model (~25 point/m$^2$, ~20 cm spacing, Table 1). This spatial distribution is consistent with the different data acquisition methods—fixed scanner positions with the TLS systems producing a $1/r^2$ falloff in point spacing for each scan with overlaps in the scans' increasing point density vs. a range of look directions for the airplane data with a 3D image acquisition geometry.

For the drone data in Surprise Canyon, we illustrate the range of point densities afforded by different processing schemes. Figure 5c,d illustrate a reason for the improvement in visualization provided by "ultra-high-density" Metashape models with an increase in point density ~2.5 times the point density of the "high" density option (200 vs. 500 pts/m$^2$, Table 1), an estimate consistent with Agisoft's documentation (www.agisoft.com, accessed on 29 May 2023). Similarly, point densities from Pix4D processing illustrate why our qualitative observation of generic Pix4D models appears to be of lower resolution because in Figure 5e, the point densities in the Pix4D models are dominantly in the 20–60 pt/m$^2$ range vs. 180–220 pt/m$^2$ in the high-density Metashape model (Figure 5c). Even the high-density option in Pix4D (Figure 5f) produces a more sparse point cloud than the high-density option in Metashape, albeit with a more geographically dispersed density in the Pix4D model and with higher densities in parts of the model. Note that the high-density options in both Pix4D and Metashape take comparable processing times. Thus, our analysis here suggests the programs produce complementary results when data are processed to high-density options, whereas there is no Pix4D processing comparable to the "ultra-high" option in Metashape.

Finally, Figure 6 illustrates a comparison of point densities from an SM model developed from an autonomous DJI Mavic drone flight flown at 100 m vs. a USGS LiDAR survey collected over the same area as part of the USGS national LiDAR data acquisition program (https://www.usgs.gov/3d-elevation-program, accessed on 29 May 2023). The survey area is in southern New Mexico, just west of Kilbourne Hole and represents a low-relief site (total relief less than 30 m) in volcanic terrain. As such, this comparison serves as a general reference for large parts of the United States where relief is low and LiDAR data of this type is, or will be, available. This analysis shows an expected result based on USGS metadata for the survey. That is, the survey was flown with NS flight lines with an overlap of ~25%, producing a striped resolution (Figure 6c) with a background point density of ~2.5 points/m$^2$ and approximately double density in the overlap zones (~5 points/m$^2$). These point densities equate to a point spacing of ~0.5–1 m for the model. The SM model, in contrast, has ~5 times the point density of the LiDAR data (Figure 6a and Table 1) and the point densities are relatively evenly distributed across the model. Note that the point density of the SM model is approximately a minimum density for this drone because the drone was flown near the legal maximum elevation for drone operations in the United States and data were processed to the "high" point density option. Thus, had we flown at a lower elevation, processed the data to the "ultra-high-resolution" option in Metashape, or both, we would have increased point densities well above the measured densities here. Moreover, note an important distinction in any visualization obtained from these data: Not only are the SM data a higher resolution than the LiDAR data, but the SM data are also a fully colored point cloud, while the LiDAR data carry no color information. Thus, although the LiDAR data can be used to produce a colored visualization through an image drape, or other methods, none of those steps are needed to use the SM model directly as a visualization.

### 3.2. Cloud–Cloud Distance Measurements

#### 3.2.1. Data Processing Procedure

In CloudCompare and PointStudio, we first compared the LiDAR and SM models visually, simply by viewing both clouds simultaneously, to look for obvious mismatches and discrepancies between model pairs. The LiDAR model is assumed to be the most

accurate, so we compared the SM models to the LiDAR in all cases. In all cases, model offsets were not obvious when viewing the entire model, and in most cases, inspection showed the main differences were in the z (vertical) dimension, i.e., one cloud was above the other, covering most of the underlying model in the visualization. In all cases, however, this visual inspection is not quantitative, and thus, we used the "cloud–cloud distance" function in CloudCompare to measure the actual mismatch. This function has three levels of analysis, and in this study, we used the least squares plane method for full point clouds because that method requires lower computation times. Specifically, the large point clouds analyzed here (>40 million points) required computation times of 3–10 h on our 8 and 12 core Windows machines, and other methods took days. Because of this issue, for most of our drone-based models with >80–100 million points, we decimated the point cloud to point spacings of 20 to 50 cm, which sped up operations and allowed use of the more sophisticated algorithms in CloudCompare. The results of the cloud–cloud distance measurements showed only minor differences among the three CloudCompare methods, presumably due to differences in resolution (e.g., point density) among the data sets.

We present the results of the CloudCompare analyses in two types of graphics: (1) an error cloud, contoured by color ramp tied to the mismatch between clouds and (2) a histogram of the total number of points for each range of differences between the LiDAR and each camera-based model. The histogram color scheme is used to color the error cloud. Note that we did not use the same range for the color scheme in different models because the parameter varied sufficiently so that many models would have been monochrome, obscuring details of model mismatch. In all cases, the overlap between our models was incomplete, and thus, we used the initial cloud–cloud distances estimates in the function to limit the maximum search distance for model comparison. This value was less than 20–50 m in all our data. After the computation was complete, we then produced a histogram display for the analysis, which invariably showed a Gaussian tail to larger errors. To refine the visualization, we visually estimated the position where this tail was approaching zero and used the "min-max" function in CloudCompare to split the error cloud into far offset vs. close offset components. This function recolors the error cloud to improve the error visualization and removes the larger offsets that are produced by data outside the overlap of the models, large holes in one of the models, or both. CloudCompare also allows the cloud–cloud distance function to estimate the X, Y, and Z components of the mismatch, and we used this function extensively where the discrepancy was not clear from visual inspection.

### 3.2.2. Analysis of Aircraft Imaging SM vs. LiDAR

For this study, we used a subset of our aircraft-acquired imagery over areas that had been covered in our 2013 LiDAR data. In Pleasant Canyon, the area was imaged with only a single camera (Nikon Df) in a single pass looking up the canyon as the aircraft flew along the mountain front. In Surprise Canyon (Figure 7), both cameras were used with three passes along the mountain front and a fourth pass downcanyon to form a grid pattern.

Because the Pleasant Canyon data were acquired on a single pass, data processing was straightforward. The data were processed unreferenced in both Pix4D and Metashape, then georeferenced using ground control points (GCP) obtained from Google Earth as described above. Given the simplicity in the data acquisition scheme for the Pleasant Canyon data, the SM-LiDAR comparison indicates a close correlation between the models. Specifically, the histogram for cloud–cloud distance (Figure 8) shows that virtually all of the SM model corresponds within 5 m, and most of the data are within 2 m. Component analysis as well as the error cloud (Table 2) show that nearly all the error is in Z with the largest errors on the south wall of the canyon where camera angles were poor for imaging. This range of error is comparable or better than the size of the GCPs used to reference the model and suggest the Google Earth georeferencing method was very successful for these data.

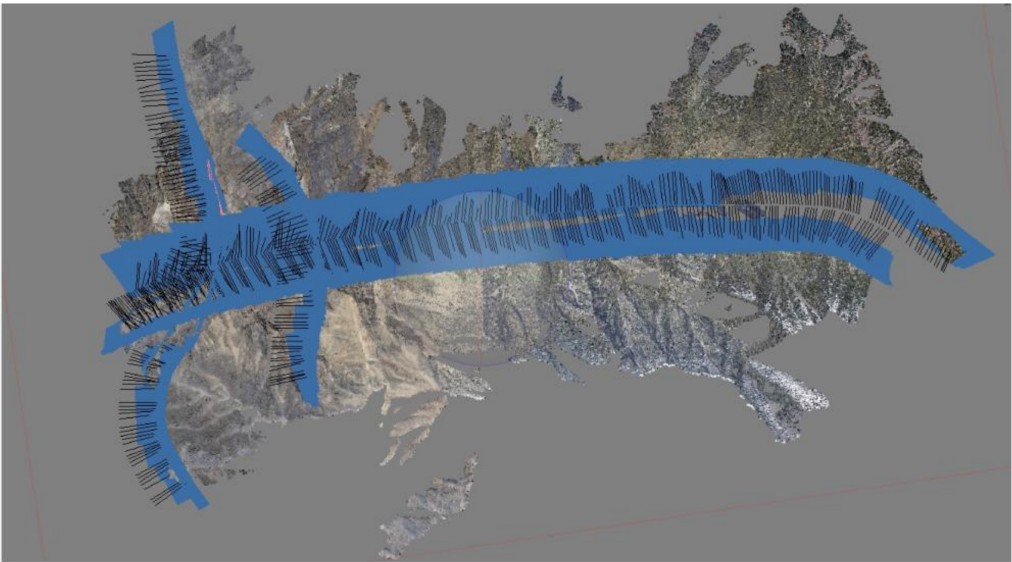

a.

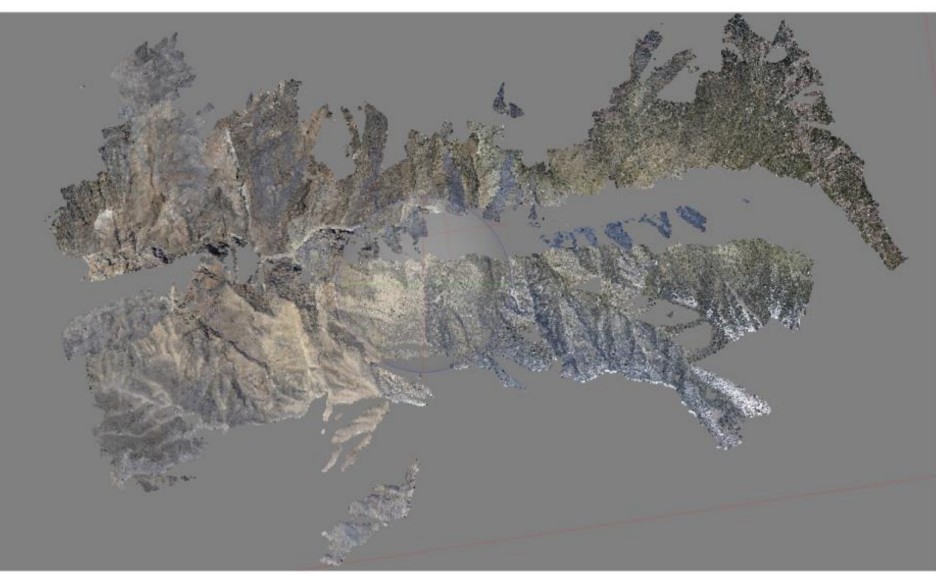

b.

**Figure 7.** Map views of the SM model developed from the images acquired in the charter airplane flight. (**a**) shows the final camera positions after georeferencing and camera merges and (**b**) shows a simple visualization of the point cloud alone.

The Surprise Canyon data presented a greater challenge with two different cameras and a cross-line geometry (Figure 7). We experimented with different techniques processing these data (Table 1): (1) an accelerated processing scheme (Model 1) to simulate a scenario that might be conducted while in the field, and (2) an in-depth data processing scheme (Model 2) that included image processing to improve the overall visualization. We estimate model 1 could be accomplished within 24 h, whereas the second would require at least a week of dedicated processing time.

Figure 9 and Table 2 summarize the results of these approaches for these data. In the accelerated processing model (Model 1, Table 2) the results are disappointing with a significant percentage of the model diverging from the reference LiDAR model by more than 10 m. This suggests this type of procedure should be avoided unless field processing is essential, errors of 10s of meters are allowable, or both. In contrast, Model 2 led to results comparable to the Pleasant Canyon data (Figure 9). Specifically, virtually all of the data

are within 5 m of the LiDAR reference with most of the data within 2–3 m. Component analysis (Table 2) shows that, as with the Pleasant Canyon model, the error is largely in Z, and on the error cloud, that error is spatially distributed with the largest errors on the south side of the canyon and west-facing slopes on the north side of the canyon (Figure 9). Given this spatial distribution and the dominance of Z in the error, we suggest that, as with Pleasant Canyon, this error results from poor elevation control on GCPs provided by the low-resolution DEM of Google Earth. Note that this conclusion is not surprising given the terrain where the scale of topographic features is much smaller than the resolution of the DEM used in Google Earth in this area.

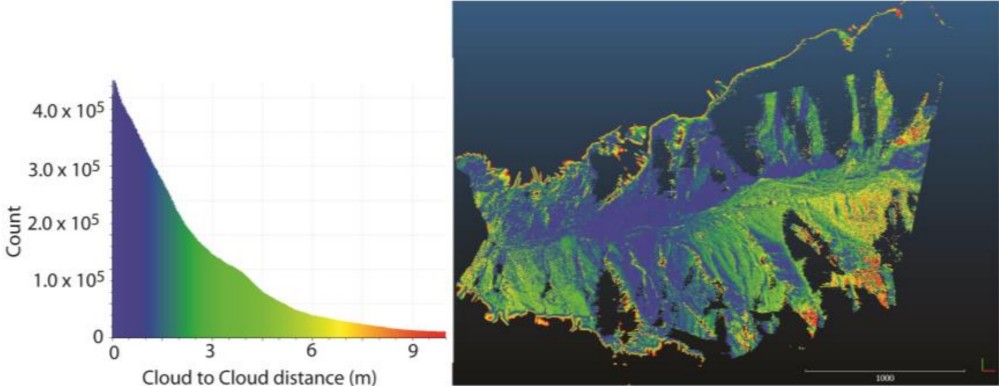

**Figure 8.** Absolute distances between LiDAR point cloud and SM point cloud model developed from images acquired in the chartered airplane flight. Histogram (**left**) shows point totals vs. distance and the color ramp is used in the error cloud (**right**), i.e., blue colors are in close correspondence between models, whereas warm colors are large mismatches.

### 3.2.3. Analysis of Drone Imaging SM vs. LiDAR

There are many permutations possible in the analysis of our data. In Pleasant Canyon, we conducted multiple flights at different times of the day and different times of the year with both drones. In Surprise Canyon, we conducted fewer flights because of greater logistical challenges, but that area serves as an important comparison because the canyon is narrower and deeper than Pleasant Canyon. We conducted a number of experiments (Table 2), but here we focus on four subsets of the data: (1) models developed from a single flight, or a pair of successive flights, with a single drone; (2) models developed from an autonomous flight flown above the canyon; (3) models developed from merging several independent flights into a single model; and (4) models where the flight plan led to poor geometry of the imaging array vs. flights with a good 3D imaging array. Each subset has distinctive features.

For subset 1, Figure 10 shows one family of results. The data in Figure 10 were from flights flown the same day in March 2022 but approximately 4 h apart. Two observations are relevant in Figure 10. First, the afternoon flights (Figure 10, upper left) show a systematic error of ~3 m shown as a peak on the histogram. Visual inspection of the two models and component analysis (Table 2) reveals that this systematic error is primarily in Z because the drone model appears below the LiDAR model. Moreover, the conspicuous blue stripes on the error cloud (Figure 10, upper right) correspond to vertical to near vertical cliffs: sites where a vertical shift would minimize the calculated misfit in the method used here. This interpretation is confirmed in component analysis (Table 2).

**Table 2.** Offset of SM models vs. LiDAR.

| Location | Acquisition Method | Georeference Method | Absolute Distance (Range) | Absolute Distance (Histogram Peak Showing Systematic Error) | X (East) Component of Offset | Y (North) Component of Offset | Z (Up) Component of Offset |
|---|---|---|---|---|---|---|---|
| Pleasant Canyon | Handheld camera from chartered airplane | Georeferenced w/Google Earth | 0–9 m. Poorest alignment where camera angles were unfavorable | none | <1 m | <1 m | <1 m, up to 3 m where camera angles were unfavorable |
| Surprise Canyon, Model 1 | Handheld camera from chartered airplane | Quick georeferenced w/Google Earth | 0–60 m. Poorest alignment at east end of model (bad GCP?) | 15–30 m | nd | nd | nd |
| Surprise Canyon, Model 2 | Handheld camera from chartered airplane | Extensive image processing and georeferenced w/Google Earth | 0–5 m. Variance in offsets across canyon and variation with look direction | <1 m | <1 m | <1 m | <1 m |
| Pleasant Canyon | single drone flight, DJI Mavic Pro | Standard differential GNSS camera positions | 0–9 m, core data 0–4 m | 4 m | <1 m | <1 m | 3–4 m |
| Pleasant Canyon | single drone flight, DJI Mavic Pro | Standard differential GNSS camera positions | 0–3 m | None, increase error at ridgetop due to sparse LiDAR? | <1 m | <1 m | <1 m |
| Pleasant Canyon | Autonomous drone flight, DJI Mavic Pro | Standard differential GNSS camera positions | 0–5 m | ~2 m | <1 m | ~1 m | −2 m |
| Pleasant Canyon | DJI mini 2, single flight | Standard differential GNSS camera positions | 0–8 m | ~4 m | <1 m | <1 m | ~4 m |
| Pleasant Canyon | DJI Mini1, 5 close range flts merged | Standard differential GNSS camera positions | 0–2 m in core model area, larger outside imaging array | <1 m | nd | nd | nd |

**Table 2.** *Cont.*

| Location | Acquisition Method | Georeference Method | Absolute Distance (Range) | Absolute Distance (Histogram Peak Showing Systematic Error) | X (East) Component of Offset | Y (North) Component of Offset | Z (Up) Component of Offset |
|---|---|---|---|---|---|---|---|
| Pleasant Canyon | DJI Mavic Pro, 4 flights, different days, processed in Metashape | Standard differential GNSS camera positions | 0–8 m | Peak at 0, secondary peak at 3 m | <1 m | Peak at 0, tail to +5 at sites close to scanner | Peak at 0, tail to −5 at sites close to scanner |
| Surprise Canyon, north bank | DJI Mavic Pro, single flight, small variation in Z in flight plan | Standard differential GNSS camera positions | 0–10 m, error systematic across model showing rigid body rotation | ~4 m | <1 m | Peak at 2m, error varies across model showing rigid body rotation | Peak at ~−3 m |
| Surprise Canyon, sharp turn in canyon | DJI Mavic Pro, single flight | Standard differential GNSS camera positions | 1–4 m | <1 m | <1 m | <1 m | <1 m |

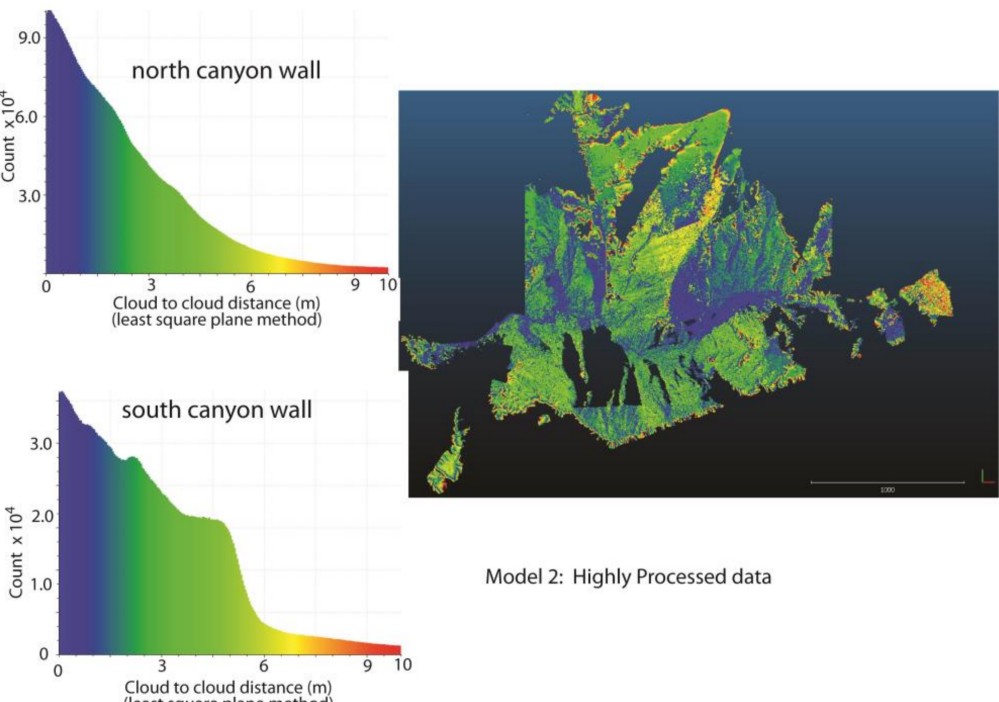

**Figure 9.** Absolute distances between LiDAR point cloud and SM point cloud models developed from images acquired in the chartered airplane flight in Surprise Canyon with extensive data processing of Model 2. Histograms (**left**) show point totals vs. distance, and the color ramp is used in the error cloud (**right**), i.e., blue colors are in close correspondence between models, whereas warm colors are large mismatches. Note the difference in color schemes for the south vs. north side of the canyon (histograms on left) merged into a single error cloud.

In contrast, the flight that same morning (lower half Figure 10) shows no comparable misfit with a near Gaussian curve in the histogram plot (Figure 10, lower left), indicating nearly all of the data are within 3 m. Indeed, all of the misfits >3 m are either located around holes in the LiDAR data (reddish rings surrounding black spots) or are on the fringes of the LiDAR model. Importantly, there is no systematic spatial distribution of the error visible on the error cloud (Figure 10, lower right) other than a slightly larger error toward the top of the cliff face, which likely reflects the fact that the drone was farther from the ridgetop due to the flight plan, the LiDAR data are sparse, or both. In other flights in Pleasant Canyon, we saw similar discrepancies from flight to flight (Table 2). Had we placed ground control points in these scenes, it is likely we could have avoided issues such as that in Figure 10, but in this site, that would have been virtually impossible on the steep cliffs that were imaged. Because these errors in Z are common within our drone data, we tentatively conclude that these errors arise from a well-known problem in GNSS data, the lower precision in Z vs. horizontal (https://www.gps.gov/technical/ps/2008-WAAS-performance-standard.pdf, accessed on 29 May 2023).

For subset 2, Figure 11 shows comparisons of the SM model for an autonomous flight flown from the ridge north of Pleasant Canyon at 90 m above the ridgetop. Note that in the drone deploy flight plan, the data were flown as a series of ~NS flight lines plus a perimeter flight where the camera is pointed inward toward the center of the model. Thus, on the cliff face where the model overlaps with the LiDAR (Figure 11), the images are primarily nadir (vertical) images with a few images looking northward, toward the cliff at a 45 degree incidence. The comparison (Figure 11) shows small systematic errors in both Z (up) and Y (north) components but negligible errors in X (east). As noted above, errors in Z are

common in these data, likely due to GNSS imprecision, but the origin of systematic errors in Y are less obvious. The systematic shift is most conspicuous in a comparison of an east view vs. a west view of the model (Figure 11), with the yellow tones on the west view illustrating the shift on cliffs that face nearly south, and green tones (smaller mismatch) on an east view showing cliffs that face southwest. Although more data are needed, our tentative interpretation of this observation is that this shift originates from the limited look directions in the autonomous flight together with GNSS error. That is, the dominance of nadir images with only a few images looking north and down may limit the accuracy of the model, particularly in Y, which is affected by depth calculations in the model. The larger errors in Y in west views (Figure 11g) are not due to errors in X because X is nearly perfectly aligned, but instead, results from cliff face geometry with south-facing cliffs showing larger apparent errors in Y vs. northwest-facing cliffs seen in the east view (Figure 11f).

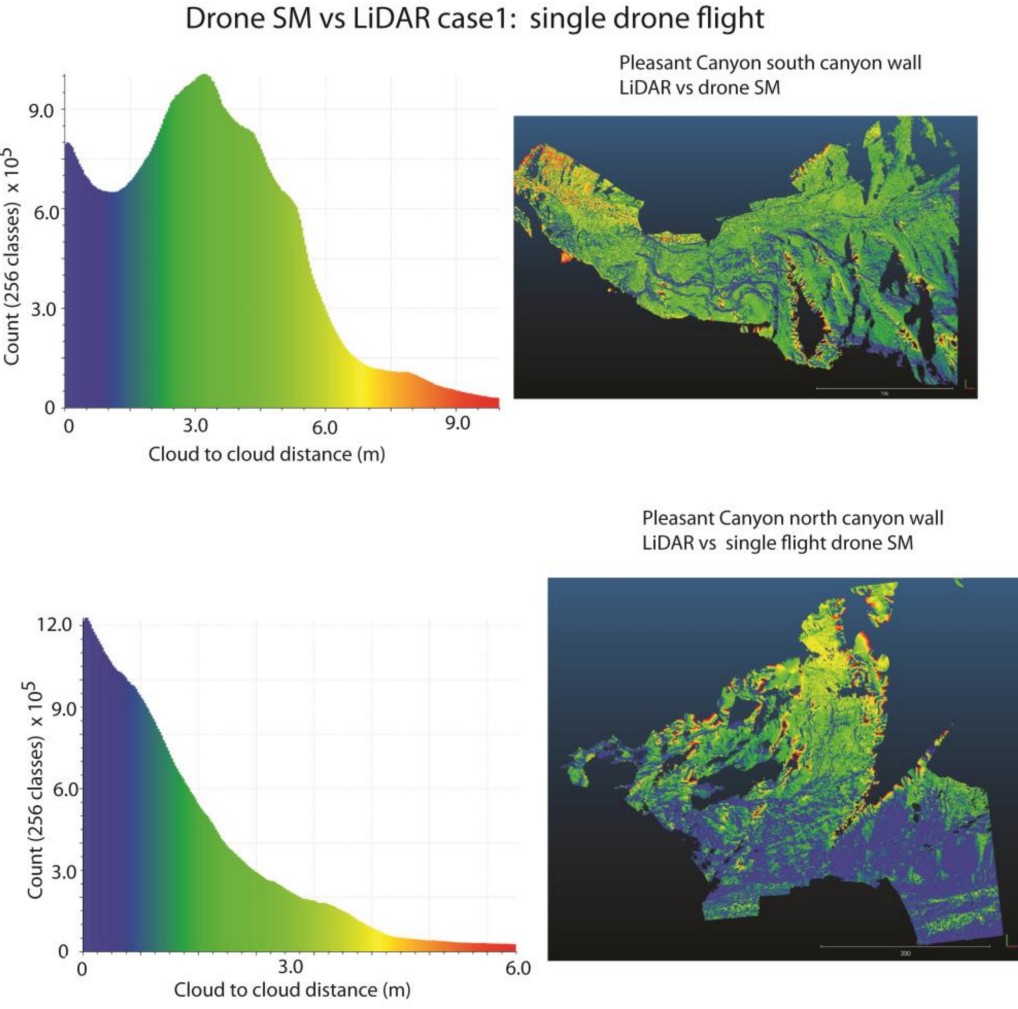

**Figure 10.** Absolute distances between LiDAR point cloud and SM point cloud models developed from images acquired in single drone flights in Pleasant Canyon. Upper Figure shows results from a pair of consecutive flights in the afternoon along the south canyon wall, whereas the lower figure is from a single flight along the north canyon wall, taken the morning of the same day. Histograms (**left**) show point totals vs. distance, and the color ramp is used in the error cloud (**right**), i.e., blue colors are in close correspondence between models, whereas warm colors are large mismatches.

## Autonomous (vertical imaging) drone flight vs LiDAR, case 2

**Figure 11.** Cloud–cloud distance estimates as components showing discrepancies between the SM model developed from the autonomous flight over the north wall of Pleasant Canyon vs. the LiDAR model. Histograms on left (**a**,**c**,**e**) show cloud–cloud distances on a decimated point cloud using the quadric method in CloudCompare with error clouds on the right shaded to correspond to the histogram. Upper figures (**b**,**d**) are map views of the error cloud, whereas the lower figures of the Y component show different spatial views. (**f**) is a view east, upcanyon, whereas (**g**) is a view downcanyon to the west and (**h**) is a view to the north. Note the larges mismatch in Y (yellows) is for south-facing cliffs. See text for discussion.

Subset 3 provides some of the most intriguing results. The merging of images from individual drone flights into a single model should improve model accuracy because, in theory, the more look directions, the better the model. However, systematic errors in georeferencing that vary among individual flights (e.g., Figures 10 and 11) could cause problems such as image distortion or rigid body rotation in the merged model. Figure 12 shows a family of results in merging data. Here, five flights taken over a period of four days during three separate field excursions were merged into a single model, processed with all images used for the model as a single data set. This data merge was aided by the

fact that all flights were made at approximately midday, albeit with different sun angles in different seasons. Nonetheless, because the imaged cliff was more or less in full sun during all of the imaging, changing shadows had minimal effects. Thus, both in Pix4D and Metashape, the data were easily combined into a single model. In the merged model (Figure 12), the correspondence with the LiDAR data is variable across the model with variations primarily in the Z (up—middle section, Figure 12) and Y (North—lower section, Figure 12) components with little variation in X (east—not shown). In line with the analysis of individual flights from which the model was built (e.g., Figures 10 and 11), the Y and Z component vary the most (Figure 12) with a close match in X (not shown). Note that the absolute error (top, Figure 12) varies across the model with the largest errors near the base of the slope, where the LiDAR data are most dense but the SM model is most poorly constrained by imaging geometry. In the histogram for the Z offset, much of the data are well aligned (near 0) but tail off toward negative values as large as ~−5 m (middle histogram, Figure 12). This offset is clearly primarily at the base of the slope, as seen in an east-view visualization of the LIDAR RGB point cloud together with the Z error cloud (middle right, Figure 12) where the LiDAR model lies above the SM error cloud. The Y offset shows a similar offset, with the largest error at the base of the slope (bottom, Figure 12). The component analysis, in this case, clearly explains the absolute offset at the base of the slope as a combination of Z and Y offsets—the SM model below and slightly north of the LiDAR model at the base of the cliff. We suggest this observation is explained by the flight plans of the SM data used to construct the model. Specifically, all the drone camera positions were high above the valley floor and focused on the upper cliffs, including an autonomous flight above the cliffs, indicating the data are highest resolution on the upper parts of the cliffs. In contrast, the LiDAR data were acquired on the valley floor, looking up at the cliffs and the highest resolution is on the valley floor. Thus, the drone data contain a mismatch because of the poorer resolution of the SM model on the valley floor.

Subset 4 is best illustrated in Surprise Canyon. This canyon affords a somewhat different perspective in drone comparisons to our LiDAR data because the canyon is significantly narrower, and deeper, than Pleasant Canyon. This distinction affects the results of both the LiDAR and drone data because: (1) the narrow canyon was a handicap to the TLS survey because upper cliffs could not easily be viewed to make a model, producing a model with the highest resolution toward the valley floor and poor resolution on upper slopes; and (2) conversely, the drone imagery tended to be flown well above the valley floor for flight safety, producing a model focused on the upper parts of the canyon walls. In addition, the narrow canyon led to known GNSS location errors on the canyon floor including real-time positioning with obvious mislocations, drone GNSS systems having problems defining a "home" position, and warnings from drone software about location problems. Thus, we knew that GNSS data used in camera locations could have significant errors, making these data an important test case.

Figure 13a–d illustrate a model with a serious error that was produced by the combined effects of a defective flight plan and GNSS errors. In this case, the error appears as both shifts in histogram peaks (Figure 13b) from the expected (0) position as well as a clear geographic variation in the error with the largest discrepancies near the top of the cliff (Figure 13a). When analyzed in terms of components (Table 2), the largest error is in Z (peak at ~3 m), but there is also significant error in Y (north) with virtually no error in X. Together, these observations indicate this model has a systematic error with the model too low and too far north near the upper parts of the cliff, but a pattern that is systematic across the model. From our experience with ground-based models [3], this pattern indicates clear evidence of a rigid body rotation in the model, with a tilt to the north. We interpret this result as a quirk of the imaging scheme (Figure 13c,d) used in the acquisition of this model where, because of the tight canyon, the flights were flown relatively high, but through a limited altitude range. Thus, we introduced a geometry problem in the data akin to the problems with ground-based models recognized by Brush et al. [3] where the imaging array was relatively linear, allowing rigid body rotations in the absence of ground control.

This problem was likely compounded by poor GNSS locations. None of our other flights in Surprise Canyon suffered from this problem because of different flight plans, although, as with Pleasant Canyon, single flights showed variable errors in both Z and Y (Table 2).

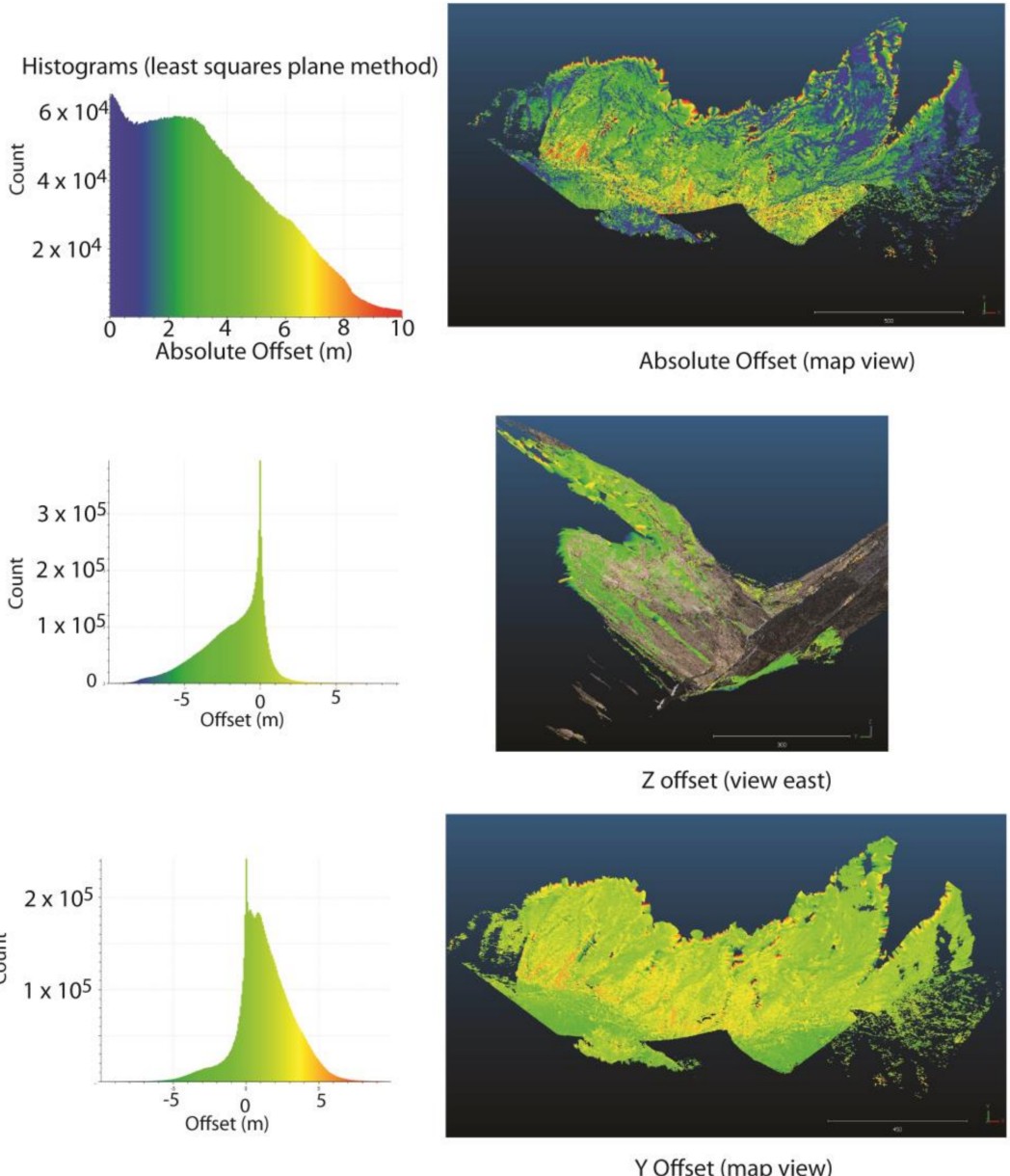

**Figure 12.** SM models produced by merging images from 5 Mavic pro drone flights vs. LiDAR reference. Error clouds on the right with respective histograms showing color ramp on the left. Lower two pairs of figures are components of the total offset shown in the upper pair of figures. See text for discussion.

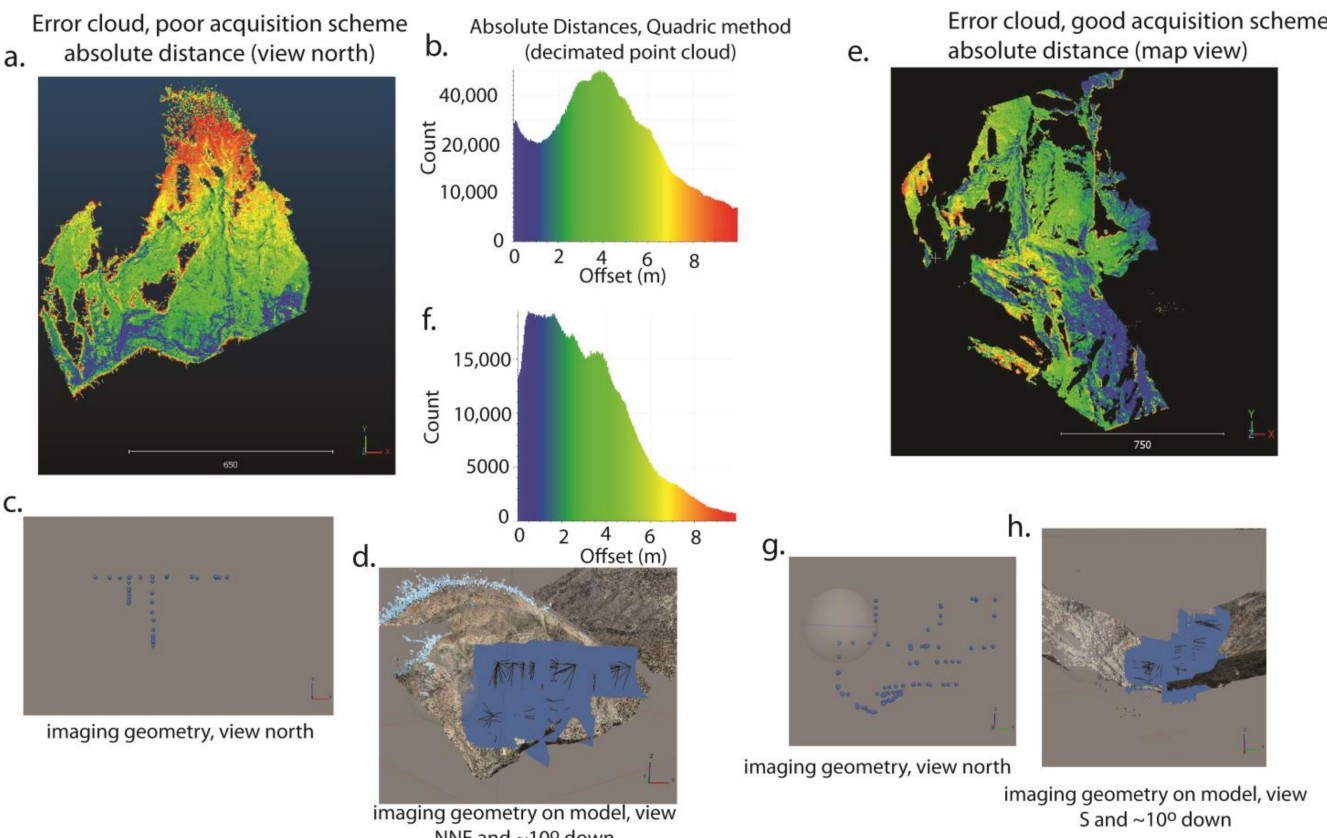

**Figure 13.** Illustration of errors introduced by acquisition geometry. (**a–d**) show data from a drone flight with poor acquisition geometry in lower Surprise Canyon. (**a,b**) show the absolute error cloud and histogram, respectively, of offset relative to the LiDAR data. Note the systematic increase in error across the model indicative of a rigid body rotation. (**c,d**) illustrate the poor acquisition geometry with a relatively linear array aside from imaging while the drone was ascending and descending, allowing a rigid body rotation of the model. In contrast, the model on the right is just upcanyon from the other model and shows offsets from the LiDAR model in the range of GNSS errors. (**e,f**) show the error cloud and histogram, respectively, for these data relative to LiDAR, with all data other than fringes of the model within ~4m and nearly all variance in the vertical (not shown). (**g,h**) illustrate the acquisition geometry that forms a 3D grid pattern, which eliminates the rigid body rotation problem in (**a–d**).

In contrast, one pair of flights flown in Surprise Canyon showed a close correspondence with the LiDAR data (Figure 13h). In this model, all three components show a strong peak within 1 m of the LiDAR (Table 2). We suggest this model produced a close correlation to the LiDAR because of the combined effects of a good flight plan and a geographic quirk of the canyon geometry that aided both the TLS acquisition and drone imaging. Specifically, at this site, the canyon deviates from a relatively linear geometry with a sharp jog in the canyon (Figure 13e–h). For the TLS data the geometry allowed a greater range of look direction as well as greater overlap among scans, increasing the density of the LiDAR point cloud, particularly higher on the slopes. Similarly, this geometry allowed the drones' GNSS to receive data from satellites low on the north and south horizons: satellites blocked in the deep, EW sections of the canyon, affording higher precision image locations that improved model accuracy. More importantly, however, this model was obtained with a 3D flight geometry (Figure 13g,h) that eliminated the potential for rigid body rotation similar to that seen in Figure 13a, leading to a closely correlated model.

## 4. Discussion

In this project we compared a wide variety of data acquisition methods to determine how these methods could be used to assist and improve geological field observations. There was no clear standout method identified; the data suggest that all the methods could significantly improve current field data acquisition with adequate planning. What appears to be most important is to plan for the type and quality of data needed for a given project. The major considerations we feel are important for a geological field study are the desired resolution and cost in terms of time, implementation, and money.

### 4.1. Resolution and Scale

Table 1 summarizes the results of our study in terms of the point density from the study areas. Point density is a measure of resolution and, if we assume the points are uniformly distributed, then we can obtain an estimate of the size or scale of objects we should be able to resolve in our models. All of the methods we used had an "average" resolution of less than 1 m and most were of the order of a few cms. However, when we look at the distribution of the point densities in our data (https://www.youtube.com/watch?v=Ajk34HKzGDo accessed on 29 May 2023 and Figures 2–6), there is significant variation among the methods. The primary factor in this is the distance (r) of the camera or scanner from the target area. That is, by simple geometry, spherical spreading of the light waves over the distance, r, increases as a function of $r^3$, which indicates that the closer the scanner or camera is to the objective, the better the resolution will be. We observed a comparable decrease in resolution with distance in the TLS data because the scanner was located at the base of a steep-walled canyon and took data systematically up the canyon walls so that the distance between the scanner and the objective increased with the height of the wall above the scanner. In contrast, the LiDAR data from near Kilbourne Hole, NM show uniform point densities aside from swath overlap (Figure 6) consistent with a method where the spherical spreading geometry has an insignificant impact on resolution because the scanner is located on a moving platform.

In contrast, all our SM data show relatively uniform point densities, in line with the SM method where the Multiview stereo step fills in areas between key points. This is perhaps most prominent in the autonomous drone flight at Kilbourne Hole, NM (Figure 6), because the land surface was fairly flat and the drone flew at a constant elevation so the distance did not change significantly. Accordingly, variations in point density are insignificant. In the steep terrain of the Panamint Mountains, however, the point density was most affected by flight parameters. We used the DJI Mini for close photography because it was easy to maneuver but hard to track at any great distance from the operator. Thus, all of our DJI Mini models were made from close-range images leading to the highest resolution models (Figures 2 and 3 and Table 1). Point density dropped off with distance as shown by DJI Mavic models (Figure 3) vs. Mini, and even farther in the airplane models (Figure 5); a result expected from simple geometry and optics.

Another factor in the resolution is the camera used for the images. A poor-quality camera or inadequate consideration of lighting conditions can make the best photogrammetry survey a waste of time. It is possible to post process images after data collection, but there are limits on what can be done. We also looked at the data processing steps in a couple of places and recognized there are options to increase the number of points added to a model after the Structure-from-Motion step that can increase the point density. We show one example in Table 1 where we used different software and/or processing options to improve the point density of our Surprise Canyon model. The Metashape ultra-high-density option gave the highest resolution but also took significantly more time than any of the other methods. Pix4D typically built the fastest model but with lower resolution.

Point density is also dependent on the amount of overlap between images and/or scans. The greater the overlap, the higher the resolution. The simplest example is in the Kilbourne Hole LiDAR data where the USGS data show a striped pattern in the point density display because there are twice as many points in the areas where the scanner

overlapped with a previous scan (Figure 6). This is, however, more subtle in SM models. Upon visual inspection, there does not appear to be any variation in density in most SM models, yet, in detail, such variations exist (Figures 2–6). Most prominent is the Surprise Canyon scene processed in Pix4D where the point densities are highest in the areas of great image overlap (Figure 5e,f), yet, even in that scene, the visual distinction between the Pix4D and Metashape "high-density" models is indistinct, particularly compared to the "ultra-high" resolution model.

The variations that we see in the various point cloud data sets were all predictable from the geometry, data acquisition method, and camera parameters. The resolutions we see in these models (Table 1), however, provide important guidelines for the data acquisition of SM or LiDAR data to achieve desired results when applied to a field campaign. Specifically, if features to be analyzed for geometry have scales of meters, even the lowest resolution data we analyzed (Kilbourne Hole LiDAR) would be adequate to resolve geometry, particularly in a flat terrain such as that area. When features become smaller, however, more high-resolution techniques become appropriate that are largely dependent on distance to the target, e.g., airplane models with resolutions to the decimeter level vs. drone models to the cm or even mm level possible.

### 4.2. Comparison between LiDAR and SM Photogrammetry Model Scale and Absolute Registration

In this study, we used the LiDAR data from our areas as reference to check the validity of our SM models by comparing them directly and to measure how much our model positions differed from the LiDAR positions. The result of our analysis here suggests strongly that SM models are very accurate renderings of the terrain that very faithfully reconstruct overall terrain geometry with the primary errors occurring as rigid body translations, rotations, or both, relative to the LiDAR reference. That is, all of our models are scaled precisely to the LiDAR reference, but the SM models are commonly shifted 2–3 m from the LiDAR reference with some models aligned perfectly to the LiDAR.

The results reported here indicate that drone-based SM models georeferenced by camera positions based on standard differential GNSS systems are located within the well-known error range for differential GNSS. That is, with one exception (Figure 13), all of our models were referenced to within 5 m vertical and 2–3 m horizontally with most referenced better than this range. We conclude that the main source of error in models georeferenced by this method is the random error in the accuracy of the GNSS positioning of the cameras. In most cases, the vertical error dominates, as expected, but in other cases, we see shifts that are influenced by the unique issues of GNSS positioning in a canyon that limits the GNSS system's ability to acquire satellites from low-angle positions, imprecision in depth calculations due to imaging array geometry, or both.

In one case (Figure 13), we recognized a larger error that can be attributed to a rigid body rotation similar to the rotations Brush et al. [3] experienced in ground-based data acquisition systems. In this case, the imaging array geometry was almost certainly at fault because it was nearly linear (constant elevation) aside from images acquired while the drone was ascending or descending to the flight height (Figure 13c,d). Had we reflown this flight, as with other flights flown in a vertical grid pattern (e.g., Figure 13g,h), we are certain this error would have disappeared. Thus, future studies should all recognize the pitfall of improper data acquisition. Note that this conclusion is important because this problem would have been invisible had we not had the LiDAR reference model.

Our data collected from a manned aircraft provide an important test of an alternative method where a model is developed for unreferenced cameras and then georeferenced using natural objects as GPCs referenced by Google Earth imagery and tools in Google Earth. Figures 7–9 show that with careful image processing and SM processing, an un-referenced model can be referenced as well as, or even better than, drone imagery where only camera positions with standard differential GNSS errors are used for the georeference. The Surprise Canyon model is particularly important in this context because it produces a dual problem of unreferenced images and images acquired from two different cameras

looking in different directions (Figure 7), yet the final model is closely correlated to the LiDAR reference (Figure 9).

A question that arises from this observation is why these models, whether aligned by Google Earth GCPs or GNSS camera positions, are aligned so well, given all the assumptions in the method? The answer to that question will require more experiments of the type we conducted here, but we suggest that the main reason these methods worked so well is scale. Specifically, the models developed from these data all cover significant areas, comparable to map scales of traditional 2D field geology, i.e., models from 2–20 km across. At these scales, errors in GCP positions (or camera positions) in the range of 1–5 m are a tiny fraction of the model size. For example, even a 5 m error across a 2 km model is only 0.25% of the model dimensions and a 1 m error across 20 km is only 0.005% of the model dimensions. That arithmetic illustrates one reason why a simple observation from our data, that all of our models were scaled within error of the LiDAR model, is seen, i.e., scaling errors are trivial as long as model size is >> the GCP (or camera) positional error. Similarly, when dispersed across the model as GCPs or an imaging array that is at least 2D (nonlinear), rigid body rotations in the model are negligible. Thus, the SM models are closely aligned to the LiDAR reference as long as the SM depth calculations are accurate, which they appear to be in our data. Note that this tentative conclusion is supported by work at virtual outcrop scales, e.g., [19], where the use of GCPs located by conventional GNSS led to small rigid body rotations, scaling issues, and even distortions. In that case, however, the GNSS error was a sizeable fraction of the model size (5–50 m) and those errors are not surprising [19].

*4.3. Camera Quality, Functionality, and Cost*

The remaining considerations in comparing the methods are primarily a matter of practicality. The better quality the camera, the more detail and clarity the photos are likely to provide. However, camera quality is also a function of cost and, to some extent, functionality. A field geologist who is hiking in rugged terrain may not choose to carry a heavy, expensive camera because of concerns about damaging it. A small, drone-based camera is likely to be a better choice in rugged terrain because the geologist can stop and fly the drone over features that are difficult to access on foot and look at the images in the field to determine whether their quality is sufficient for the purpose of the project.

Carrying a LiDAR system in the field is generally impractical, but there are now some portable LiDAR systems available. The Apple iPhone 12 Pro and more recent models of cell phones can acquire LiDAR data and process them in the field, but the results are unimpressive with the phone processing and they have not yet been shown to work well over large distances (i.e., 10s of meters or greater). The phone LiDAR data can be downloaded and processed using the same software we used for our camera data, and the quality appears to improve significantly (Martinez, 2022). This is a method that needs more testing but does have potential as a field data collection method. Drone-based LiDAR systems are also becoming widely available (e.g., https://enterprise.dji.com/zenmuse-l1 accessed on 29 May 2023) and may ultimately resolve some of these issues. Nonetheless, these drones remain relatively expensive, are relatively heavy, and require considerably more technical expertise to operate. Thus, for now, they are likely to remain niche applications where both accuracy and visualization are critical, e.g., engineering applications.

The comparisons discussed in this paper are similar to those reported by Martinez and Serpa [6] and Martinez [13], but the scale of the two studies is very different. Martinez et al. [6] evaluated a range of methods similar to this study to determine their applicability in a very small area, 72 m$^2$, to map dinosaur footprints. The results also showed that the methods that collect very close data were better than more remote data collection. However, location errors of the order of a meter were not acceptable in that case, and none of the techniques other than LiDAR or placing GCPs in the area appear to provide sufficient accuracy, although, as with this study, the relative positions of features appeared to be preserved. One significant outcome of the Martinez study was the use of image processing to bring out footprints in the data that were not visible to the naked eye or in camera images

because of variations in the color patterns of the host rocks that obscured the images. The topography of the footprints became apparent when the true colors of the outcrop were removed and depth contouring was added; an important contrast to the application here where color visualization is critical. Thus, application remains an important concern in data acquisition.

We have not yet explored the use of advanced image processing to this extent on the large-scale areas we looked at in this study. Nonetheless, remote measurement tools have been developed in both CloudCompare and PointStudio that can be used to make remote orientation measurements (e.g., see [3] for an example of the methods), and both software packages allow routine mapping of lithologic contacts in 3D. These types of data can be used to easily extract other information such as unit thicknesses, 3D curvature, intersections, etc. We believe that even more information can be measured, observed, and preserved using 3D digital field methods and anticipate that additional image processing may further enhance these capabilities. This suggests further reasons to collect photogrammetry data in all field mapping programs, and these programs could be further enhanced by advanced sensors such as multispectral data.

*4.4. Significance for Geologic Field Studies*

Based on all the analyses here, it is clear that all the techniques evaluated can generate terrain models with resolutions at the ~1 m to cm scale over ranges of several kilometers with increasing resolution closely correlated to camera optics, camera resolution, and image array geometry. Varying these parameters produces results that are qualitatively obvious, e.g., a longer lens produces an image with higher resolution that carries through to the model, a higher resolution sensor produces greater resolution, and images closer to the object increase resolution. All these parameters have a tradeoff, however, in data acquisition that must be kept in mind, e.g., the effort of processing a data set with 10,000 images vs. 1000 is huge and unnecessary if the subject of the investigation is at a scale far larger than the resolution obtained in the 10k image model. In addition, capturing images too close in can lead to major problems in the Structure-from-Motion step of data processing because image overlaps in the data may be insufficient for the software's feature-matching algorithm, potentially generating an unusable data set when images cannot be matched ("aligned" in the terminology of Metashape software). Our data sets illustrate these issues. In parts of our airplane data set (not shown here), there was insufficient image overlap in the image sequence, requiring more elaborate processing. In drone data, this overlap issue was less common for two reasons: (1) the images are tagged with a georeference, which allows the software to refine image searches and minimize potential for error; and (2) using the 2 s interval shooting method and slow flight speeds produced enough image overlap we rarely had image correlation problems. On the resolution issue, our data show a range of results that strongly effect their use. Our LiDAR data (Figure 2) is at far too low a resolution to resolve the fine structural details we observed in Pleasant Canyon (cm scales). In contrast, all of the SM data are sufficient for this task, yet obtaining more imagery and/or higher resolution models through processing such as the "ultra" option in Metashape are unnecessary in most cases. Similarly, in a study related to this one [8], an image set acquired from a fixed-wing aircraft using a technique similar to the one we describe here was more than adequate to resolve geologic features that were at a meter scale or larger. Thus, focused drone fights were unnecessary other than in small areas where the structural complexities dropped below the resolution of the aircraft model.

This observation suggests a new paradigm for geologic field studies is now possible. Following the suggestions in Rutkofske et al. [8], conventional 2D geologic mapping, albeit digital, should be the beginning step in any field study. Where topographic relief is significant, a study might progress first to a 2.5D method using public domain GIS data with orthoimagery draped on a digital elevation model or simply Google Earth. However, where terrain is steep, the 2.5D method typically fails, with some failures invisible, e.g., [2,3], and in these cases, the high-resolution 3D methods can be truly transformative. The choice

of method, however, should depend on the scale of the problem at hand and the scale of the geologic features to be analyzed as well as logistical considerations of access. The latter is undoubtedly particularly critical in frontier areas where access is limited. In these cases, drone flights might be impossible due to time and access limitations, and in these cases, conventional manned flights would be the obvious solution. In other cases, however, where road access or reasonable access on foot is possible, a drone-based method is likely superior with the imaging details dependent on the scale of the problem. In all these cases, however, field geologists will need to learn some basic photogrammetric issues including camera parameters vs. resolution to optimize imaging studies as well as care in data acquisition; most notably, the need for significant image overlaps in an imaging sequence. Our results here (Tables 1 and 2) provide starting guidelines for these types of future studies.

Finally, we note that our experience with aircraft imaging suggests that SM applications may produce fundamental new understanding of the geology of remote areas. For example, based on field experience in remote parts of Alaska where large topographic relief in glacial valleys is the norm, and access to field sites is exclusively by aircraft, SM methods could revolutionize these field studies. In past field studies it was common to use a "fly-by" investigation of an inaccessible cliff exposure. In these fly-bys, we might photograph the site, sketch the feature, or both, while in the air, yet that procedure is neither quantitative nor accurate. For example, sketching may be crude and later interpretations of photos may be inconclusive due to image quality, lack of georeference for the image, poor recollection of the feature being observed, etc. Moreover, this activity is typically rushed because of logistical factors ranging from flight time costs to weather concerns.

Based on our experience here, a procedure that would be superior to this "quick look" approach would be to collect imagery, comparable to our aircraft flights, at a problem site such as this, and analyze the site with a 3D terrain model. In general, the data would be analyzed after field work, although it is conceivable the data could be processed at a field site with electrical power and computational power for data processing. In any case, were these data are acquired ad hoc, without a GPS-enabled camera, a researcher would need to employ a ground control technique for georeferencing. Thus, given the ubiquitous access to Google Earth, our test here is an important one to answer the question: is using natural objects as ground control, using Google Earth for reference (or GIS), a workable method?

Based on our results here, we suggest the answer is an emphatic yes, with some caveats. The principal caveat is the answer to a simple question of what level of accuracy is "good enough"? In comparison to conventional fixed-scale paper mapping where a pencil line may be 10–20 m wide, even our worst results would have an accuracy an order of magnitude better than a paper map. More importantly, however, is that in all our models, scale and shape correlated well to the LiDAR reference models, and the main errors introduced by georeferencing were rigid body translations and rotations. Thus, if absolute locations are critical, they could easily be resolved by more sophisticated georeferencing, and any geologic models developed from the data could easily be shifted to their proper position. That is, because geometry is the key for most geologic studies, not absolute positioning, these problems are not significant to most studies.

Finally, we note that this field paradigm also opens opportunities for studies that were simply impossible in conventional mapping methods because geometry could not be quantitatively evaluated. To illustrate the importance of this development, consider a structural geology example. Ramsay [24] long ago recognized that fold style, as indicated by the geometry of folded surfaces among different layers, was indicative of variability in flow between layers, e.g., flexural flow vs. flexural slip. Moreover, Ramsay [24] developed a method for quantitatively evaluating fold style that related directly to intra-layer flow using the dip isogon method. Ramsay and Hubber [25] and Ramsay and Lisle [26] showed examples of how these methods can be extended to strain variations within a fold system. Although elegant, these theories were ahead of their time because until now, little, if any, geologic mapping was sufficiently accurate to use these methods for true quantitative analysis of features 100s of m in size. With photogrammetry methods and the resolutions

we recognize here, these 3D mapping methods should now be relatively routine, and we expect a proliferation of studies of that type in the near future, entirely due to the development of the high-resolution 3D terrain models analyzed here.

*4.5. Suggestions for Further Work*

Although our work affords an assessment of the accuracy of SM data relative to LiDAR, more work is needed in the context of field geologic studies. Engineering assessments of the problem are useful but are focused on absolute accuracy as well as projects that are typically in a different scale range than the study analyzed here—10s to 100s of m in engineering vs. km scales analyzed here. Most important might be a study that analyzes the accuracy of the Multiview stereo step in SM data beyond what we show here. Specifically, our data compare a relatively low point density LiDAR data set to SM data that are at least an order of magnitude higher in point density. Thus, because the higher point density in our SM data is derived primarily from the Multiview stereo step in the data processing, the accuracy at these fine scales remains nearly unresolved by our analysis. Although the accuracy of this fine topographic texture is unimportant for most geologic analyses, it could be critical in some instances. For example, both CloudCompare and PointStudio have orientation analysis tools that allow remote measurement of planar orientations (e.g., [3,8,19]. The accuracy of these remote measurements depends on the accuracy of individual points, and if these points are closely spaced and a model has inaccuracies, the measurements could suffer.

An experiment that might be particularly fruitful to analyze our hypotheses would be to use a drone equipped with a LiDAR system and a camera to acquire a joint LiDAR/SM model of a cliff face with a size >1 km. Ideally, an imaging experiment from a manned aircraft across the same scene would be useful. An ideal experiment would also allow enough access to place high-resolution (mm level) ground control across the scene. These data could then be run through similar processing schemes as those employed here (e.g., camera reference only vs. Google Earth reference vs. high-resolution gcp, etc.).

**Author Contributions:** Conceptualization, T.L.P. and L.F.S.; Methodology, T.L.P. and L.F.S.; Writing—original draft, T.L.P. and L.F.S.; Funding acquisition, T.L.P. and L.F.S. All authors have read and agreed to the published version of the manuscript.

**Funding:** This research was funded by [National Science Foundation] grant number [EAR-2049603].

**Institutional Review Board Statement:** Not applicable.

**Informed Consent Statement:** Not applicable.

**Data Availability Statement:** Not applicable.

**Acknowledgments:** This work was supported by NSF EAR-1250388 to Pavlis and EAR-2049603 to Pavlis, Serpa, and J. Ricketts. We thank James Rutkofske and Valeria Martinez for our discussions on different data processing schemes and Rutkofske for his assistance in data acquisition during the chartered airplane flight.

**Conflicts of Interest:** The authors declare no conflict of interest. The funders had no role in the design of the study; in the collection, analyses, or interpretation of data; in the writing of the manuscript; or in the decision to publish the results.

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
