# Peer review of "Accuracy of Structure-from-Motion/Multiview Stereo Terrain Models: A Practical Assessment for Applications in Field Geology"

_geosciences, doi:10.3390/geosciences13070217_

Round 1

Reviewer 1 Report

This manuscript by Pavlis and Serpa is a valuable contribution to the literature on the various methods of remote data acquisition and processing for 3D terrain models. The authors do a thorough job of comparing methods of LiDAR data acquisition with SM methods and highlight the value of an assortment of SM approaches. While the authors do a nice job of indicating where SM methods are accurate and appropriate, they do so by comparing SM results with LiDAR results. However, the limitations of the LiDAR results are not as well explained. It would be helpful to include a short section on the known limitations of terrain models derived from LiDAR data. Some other minor issues that could be addressed are that the 2.2.3 Data Processing section incorporates some results, and the 3. Results section incorporates some elements of discussion. These issues don't noticeably detract from the flow of the paper, so the authors can decide whether to address these issues. Finally, the component of this work that will likely be most useful to readers are the results and recommendation for future applications of SM techniques. I would suggest that the authors more explicitly highlight best practices for future SM data acquisition, perhaps in a stand-alone Best Practices section, perhaps with bulleted recommendations.

Below are some line-by-line suggested edits:

Line 76: What is the "problem" that the research is analyzing?

Line 105: Explain the "2D assumption"

Line 127: I don't follow the "projected vertically" comment. Aren't these images typically draped horizontally over a pseudo 3D surface? Perhaps this just needs to be worded more clearly.

Line 149: "Finally," should be "3)", to match the previous 2 items.

Line 160: Explain how GNSS positioning greatly aids data acquisition

Line 193: Provide a reference for "the classic problem of map boundary faults"

Line 240: Explain what is meant by "an inner gorge physiography"

Line 279: "avoiding" should be "avoided"

Line 388: provide a reference for the statement about "twice the inverse of the point density" or explain the rationale.

Line 402: Explain what is meant by "that option can't change basic geometry"

Line 512: The "or both" comment is unclear

Line 558: Perhaps change "speeded" to "sped up"

Line 626: When referring to the "Z" direction, consistently use a capital "Z" throughout the manuscript (c.f. line 598 where a small z is used)

Line 645: Change "serves an" to "serves as an"

Line 650: Explain "defective flight plans"

Line 681: What are "nadir images"?

Line 769: Change "suffered this" to "suffered from this"

Line 847: Capitalize "Mini"

Line 849: I don't understand the logic behind "and thus all of our dji Mini models were the highest resolution"

Lines 874 - 883: this paragraph could be moved to a Best Practices section to highlight the recommendations more explicitly.

Line 1007: "the cameras have georeference" is unclear

Line 1081: It would help if "macroscopic scales" was defined in more detail. What are the specific scales to which you are referring?

Lines 1097-1099: Provide a reference of previous work that has used these software features.

Author Response

Reviewer 1:

This manuscript by Pavlis and Serpa is a valuable contribution to

the literature on the various methods of remote data acquisition

and processing for 3D terrain models. The authors do a thorough

job of comparing methods of LiDAR data acquisition with SM

methods and highlight the value of an assortment of SM

approaches. While the authors do a nice job of indicating where

SM methods are accurate and appropriate, they do so by

comparing SM results with LiDAR results. However, the limitations

of the LiDAR results are not as well explained. It would be helpful

to include a short section on the known limitations of terrain

models derived from LiDAR data.

Authors response:  We have done this by rewording in the sections on lidar data

Some other minor issues that

could be addressed are that the 2.2.3 Data Processing section

incorporates some results, and the 3. Results section incorporates

some elements of discussion. These issues don't noticeably

detract from the flow of the paper, so the authors can decide

whether to address these issues.

Authors response:  We understand it is important to keep results from the methods section, but we did this intentionally here because we thought a lot of our audience wouldn’t understand why some of these statements are important without knowing how they effect results.  We did do some rewording though with this thought in mind.

Finally, the component of this

work that will likely be most useful to readers are the results and

recommendation for future applications of SM techniques. I would

suggest that the authors more explicitly highlight best practices for

future SM data acquisition, perhaps in a stand-alone Best

Practices section, perhaps with bulleted recommendations.

Authors response:  Although this is a great suggestion, we are a little reluctant to push a best practices section yet, at this stage of knowledge.  We think more important are our suggestions for further work and some suggestions of ways these data should be used, and when more information like that is done, we could do a better job of suggesting best practices.  We do do some of this in the discussion and the abstract, and hopefully that will be enough for people. 

Line by line  edits

Line 76: What is the "problem" that the research is analyzing?

done

Line 105: Explain the "2D assumption"

Changed to emphasize 3D

Line 127: I don't follow the "projected vertically" comment. Aren't

these images typically draped horizontally over a pseudo 3D

surface? Perhaps this just needs to be worded more clearly.

reworded

Line 149: "Finally," should be "3)", to match the previous 2 items.

we see the  point, but writing it that way would be awkward because the list is broken up into sentences above this line.  This is really an extension of the first  item in the list, so doesn’t belong in the list.  Slight  rewording to try to help here.

Line 160: Explain how GNSS positioning greatly aids data

Acquisition

reworded

Line 193: Provide a reference for "the classic problem of map

boundary faults"

reworded

Line 240: Explain what is meant by "an inner gorge physiography"

reworded

Line 279: "avoiding" should be "avoided"

done

Line 388: provide a reference for the statement about "twice the

inverse of the point density" or explain the rationale.

Done—it is a digital signal processing concept common in geophysics, Nyquist frequency.

Line 402: Explain what is meant by "that option can't change basic

geometry"

reworded

Line 512: The "or both" comment is unclear

reworded

Line 558: Perhaps change "speeded" to "sped up"

Good point; fixed

Line 626: When referring to the "Z" direction, consistently use a

capital "Z" throughout the manuscript (c.f. line 598 where a small z

is used)

all  changed to capital z

Line 645: Change "serves an" to "serves as an"

done

Line 650: Explain "defective flight plans"

It is explained later—seems unnecessary here, but minor wordijg change

Line 681: What are "nadir images"?

Vertical images—not sure this should bechanged but added a parenthetical  expression

Line 769: Change "suffered this" to "suffered from this"

done

Line 847: Capitalize "Mini"

done

Line 849: I don't understand the logic behind "and thus all of our dji

Mini models were the highest resolution"

Good point; bad wording  fixed

Lines 874 - 883: this paragraph could be moved to a Best

Practices section to highlight the recommendations more explicitly.

THINK ABOUT THIS

Line 1007: "the cameras have georeference" is unclear

Not easy  to explain briefly because it is a subtlety of  the software.  Minor reword hopefully  address  this

Line 1081: It would help if "macroscopic scales" was defined in

more detail. What are the specific scales to which you are

referring?

reworded

Lines 1097-1099: Provide a reference of previous work that has

used these software features.

done

Reviewer 2 Report

Review of: Accuracy of Structure-from-Motion/Multiview-Stereo terrain models: A practical assessment for applications in field geology

This manuscript focuses on SfM-MVS for outcrop reconstruction, with particular focus on the accuracy of direct georeferencing (i.e., without the use of ground control points). While generally well written, the paper would benefit from being both shorter and more organized. Almost all of the figures require substantial revisions to make them publication ready. Recommendation: Major Revision. Detailed comments below and in attached PDF.

1. Introduction and Background. These two sections make up almost 200 lines of text. Suggest merging these into a single section and shortening substantially. Given that this paper does not cover any major geologic background etc., I don’t see the need for this part to be so long.

2. Methods. Generally OK. Could perhaps be a bit shorter and less conversational.

3. Results. My biggest issue with the Results is the very long section about point density (~200 lines of text, 1 table, and 4 multi-part figures). Based on the abstract, I would not expect this section to be here at all, and frankly, I’m not sure any of this is very important for the paper. I agree that point-cloud density is important for geologic interpretation but I would argue that this topic deserves its own standalone paper. The authors show that the LiDAR point-clouds are generally lower density than SfM-MVS equivalents but they don’t really address scanner type and effective range, scanning rate, field of view etc. This section, while interesting, is perhaps a bit lacking in detail, both in terms of quantifying point densities, and in terms of scanner rates, resolutions, field of views etc. The authors also make a few vague statements about the importance of point densities for structural geology (I wholeheartedly agree with this!) but they don’t provide any examples or case studies (e.g., I can see fault X is in this dataset but not the other one). Based on these comments, I suggest removing this section from the paper and attempting to write a second paper focused on this topic. Currently this section is too long and the job has only been partly done. See attached PDF for comments on figures.

4. Discussion. Nicely written. If you accept my suggestion to remove the section on point density, the discussion can be shortened a bit.

5. Figures. Generally not very easy to understand. As much as possible, it would help if equivalent histograms (e.g., point density) had the same x and y scales and the same absolute color scales were used for all datasets. This will make it much easier for the reader to understand. Same comment for cloud-to-cloud distance histograms.

The point-cloud screenshots are at times a bit difficult to interpret – particularly where 2 point-clouds are being compared (e.g., Fig. 3) but it is not apparent where they are in relation to each other, how much they overlap, or if we are looking at the two point-clouds from different directions. This just makes it really difficult for the reader to visualize, assess, or even understand what they are looking at. Again, point colors should be on the same absolute scale (e.g., Fig. 3), otherwise comparisons are difficult.

Detailed comments in the attached PDF.

N/A

Author Response

This manuscript focuses on SfM-MVS for outcrop reconstruction,

with particular focus on the accuracy of direct georeferencing (i.e.,

without the use of ground control points). While generally well

written, the paper would benefit from being both shorter and more

organized. Almost all of the figures require substantial revisions to

make them publication ready. (authors note:  see below)

Recommendation: Major Revision.

Detailed comments below and in attached PDF.

  1. Introduction and Background. These two sections make up

almost 200 lines of text. Suggest merging these into a single

section and shortening substantially. Given that this paper does

not cover any major geologic background etc., I don’t see the need

for this part to be so long.

Authors response:  This is a bit of a style comment, and generally writing style is something that, from our experience, is at authors discretion, provided the style isn’t distracting.  To us, this is just extending typical style of geoscience articles where the intro states the objective of the paper and the background discusses previous work on the subject as a background to the problem.  We thought that was what we were doing here; indeed, our intent was to keep the introduction short and use the background section to give details that, to most geoscientists, isn’t well know—the history of this kind of technology.  Maybe we were wrong on that, but we felt it was absolutely critical to give some literature review on the history of technology in geo mapping, and how it leads to this kind of system that we analyze.

With this criticism in mind though, we have made major revisions to these two sections.   The result isn’t a big difference in length though—more information in the introduction and a more brief background section.  We hope this is sufficient.

  1. Methods. Generally OK. Could perhaps be a bit shorter and

less conversational.

  1. Results. My biggest issue with the Results is the very long

section about point density (~200 lines of text, 1 table, and 4 multipart

figures). Based on the abstract, I would not expect this section

to be here at all, and frankly, I’m not sure any of this is very

important for the paper. I agree that point-cloud density is

important for geologic interpretation but I would argue that this

topic deserves its own standalone paper. The authors show that

the LiDAR point-clouds are generally lower density than SfM-MVS

equivalents but they don’t really address scanner type and

effective range, scanning rate, field of view etc. This section, while

interesting, is perhaps a bit lacking in detail, both in terms of

quantifying point densities, and in terms of scanner rates,

resolutions, field of views etc. The authors also make a few vague

statements about the importance of point densities for structural

geology (I wholeheartedly agree with this!) but they don’t provide

any examples or case studies (e.g., I can see fault X is in this

dataset but not the other one). Based on these comments, I

suggest removing this section from the paper and attempting to

write a second paper focused on this topic. Currently this section is

too long and the job has only been partly done. See attached PDF

for comments on figures.

Authors response:  This is the most critical comment raised by this reviewer and we took it to heart by doing a major revision of the section on density as well as adding sentences in the abstract and introduction describing why this analysis is critical.  We also added (rather than removed) a figure to bring home the points of this section. 

We suspect an underlying issue is our approach of using point clouds as a direct tool for visualization; something rarely  done in LiDAR  data because it is typically too sparse.  The reviewer suggests making a separate paper on this issue but we very much object to that suggestion.  We believe the two analyses together are inextricably intwined as the basis for using these data for 3D mapping and are the key to the concepts we discuss toward the end of the paper.  Many people like to keep papers simple to one concept, but in this case we believe that would greatly decrease the impact of the paper.  Nonethless, because of this criticism we have modified this part of the paper a lot, and hopefully now explains its importance better.  Perhaps most important is our new Figure 2 that shows the effect of point density on the ability to visualize features; showing a scene at the same scale with different methods, including a zoom in to show a very high density model.  We hope these  changes address this issue.

  1. Discussion. Nicely written. If you accept my suggestion to

remove the section on point density, the discussion can be

shortened a bit.

  1. Figures. Generally not very easy to understand. As much as

possible, it would help if equivalent histograms (e.g., point density)

had the same x and y scales and the same absolute color scales

were used for all datasets. This will make it much easier for the

reader to understand. Same comment for cloud-to-cloud distance

histograms.

Authors response:  This is not only difficult to impossible to do, it would actually destroy the content of most of the figures.  The purpose of these figures is to show spatial variations in point density and offset from the reference model.  If we used the same color scheme for all, many of the figures would be monochrome.  We even selectively filtered some of the results for specifically this reason—e.g. to show only low point density variations or small offsets, etc.  The histograms show the actual density variations and offset frequencies, the point cloud visualizations just emphasize where the variations occur.

Based on this comment, however, we did modify some text to hopefully clarify this more.

The point-cloud screenshots are at times a bit difficult to interpret –

particularly where 2 point-clouds are being compared (e.g., Fig. 3)

but it is not apparent where they are in relation to each other, how

much they overlap, or if we are looking at the two point-clouds

from different directions. This just makes it really difficult for the

reader to visualize, assess, or even understand what they are

looking at. Again, point colors should be on the same absolute

scale (e.g., Fig. 3), otherwise comparisons are difficult.

Authors response:  the comment above applied here too, but we have tried to address this by adding some geographic references to figures to show where feature overlap, etc.  We also plan to do a visual abstract to help with this to show animations of some of the scenes to help readers.  We realize static images of 3d models are kind of hard to see; ideally we would have liked to use 3d pdfs so people could just look at these things, but we didn’t see a way to do  that.

Line by line comments:

8 and 10: changed to precision

13 and 15:  done—we have some doubts about doing this for the abstract, but fine.

37:  done

Figure  1.  Not sure what the reviewer wants here.  “neater” can mean a lot of things.  It is a simple location map, so not sure what else is needed? 

108 and 113:  Sorry, but referencing these statements is a little like referencing Darwin for anything dealing with evolution (extreme comparison I know but).  This is essentially common knowledge in geosciences in our opinion. You can google this and get hundreds of hits.

121 and 134: done; didn’t add “spacebased” since that seems redundant with airborne (although that is above  the  atmosphere, so subtle  difference?)

164-165:  Voxel is actually a very bad term here.  A voxel is a regular grid in 3D, kind of the 3d equivalent of a DEM.  This is the 3D representation of a surface by an irregularly spaced series of points.  We could make up a term here??  But best not to proliferate  terminology if not necessary.  Reworded slightly, hopefully that helps.

230:  open topography—the  site has a very good  search engine, someone should be able to find  these data in seconds, just zoom in on the map to the  area

243-245: good point; deleted

270 and other capitalizations—fixed

427-428:  Q is where are the scanner positions, and 2b shows higher point densities

Scanner positions are  obvious on 2b; reworded the figure caption to bring  that point home.

2b—not sure  how to address this.  Figure 2b shows that only high densities are close to the scanner which is what  the statement  says.  Don’t see how rewording will clarify this; hopefullyk the rewording  about scanner positions clarifies?

Figure 2—label error cloud.  GOOD POINT!  We were fixated on the other  figures about model offsets and started calling these  error  clouds and that is definitelyi the wrong  term for these figures!  Thanks much for catching that. 

Changed to “point density  cloud”

448: added to target

Figure 3. comment 1—changing histogram scales (similar comments about figures 4 and 5).  Not only would  that be a lot of work, it would also make it much harder for someone to interpret the figure.  If you used the  same color scheme  in all the figures most  would be monochrome because of the large scale  range of the  density.  We could compress the histogram plots to make x axis the same, but that would produce a different  visualization  problem—very skinny little histograms for the low density  data.  So trying to address this  would  actually make the figure worse.

Second point, about image overlap—tried to address this by modifying the figure caption.

Figure 3-5:  we did change some of the figure captions to emphasize issues raised here to try to explain color scheme.

496: fixed

565:  fixed

Figure 9:  comment about inconsistent labels.  Our mistake.  Fixed, changed all to cloud-cloud  distance